# Hepatitis C virus exploits cyclophilin A to evade PKR

Che C Colpitts[1,2]*, Sophie Ridewood[2], Bethany Schneiderman[2†], Justin Warne[3], Keisuke Tabata[4], Caitlin F Ng[2], Ralf Bartenschlager[4,5,6], David L Selwood[7], Greg J Towers[2]*

[1]Department of Biomedical and Molecular Sciences, Queen's University, Kingston, Canada; [2]Division of Infection and Immunity, University College London, London, United Kingdom; [3]Wolfson Institute for Biomedical Research, UCL, London, United Kingdom; [4]Department of Infectious Diseases, Molecular Virology, Heidelberg University, Heidelberg, Germany; [5]Division Virus-Associated Carcinogenesis, German Cancer Research Center, Heidelberg, Germany; [6]German Center for Infection Research (DZIF), Heidelberg Partner Site, Heidelberg, Germany; [7]Department of Medicine, Imperial College London, London, United Kingdom

*For correspondence:
che.colpitts@queensu.ca (CCC);
g.towers@ucl.ac.uk (GJT)

Present address: †Department of Infectious Disease, Section of Virology, Imperial College London, London, United Kingdom

Competing interests: The authors declare that no competing interests exist.

**Abstract** Counteracting innate immunity is essential for successful viral replication. Host cyclophilins (Cyps) have been implicated in viral evasion of host antiviral responses, although the mechanisms are still unclear. Here, we show that hepatitis C virus (HCV) co-opts the host protein CypA to aid evasion of antiviral responses dependent on the effector protein kinase R (PKR). Pharmacological inhibition of CypA rescues PKR from antagonism by HCV NS5A, leading to activation of an interferon regulatory factor-1 (IRF1)-driven cell intrinsic antiviral program that inhibits viral replication. These findings further the understanding of the complexity of Cyp-virus interactions, provide mechanistic insight into the remarkably broad antiviral spectrum of Cyp inhibitors, and uncover novel aspects of PKR activity and regulation. Collectively, our study identifies a novel antiviral mechanism that harnesses cellular antiviral immunity to suppress viral replication.

## Introduction

Viruses encounter a remarkable array of intracellular antiviral defences that they must suppress or evade in order to replicate. The cyclophilin (Cyp) family of host proteins have emerged as key players at the virus-host interface. Cyclophilin A (CypA) is a cofactor for a variety of established and emerging viruses, including *Flaviviridae* such as hepatitis C virus (HCV) (*Yang et al., 2008*) and dengue virus (*Qing et al., 2009*), as well as *Coronaviridae* such as SARS coronavirus (*Pfefferle et al., 2011*). Like other Cyps, CypA has peptidyl prolyl isomerase activity, which is thought to induce conformational changes in bound target proteins (*Wang and Heitman, 2005*). Importantly, recruitment of CypA also affects protein complex formation (*Liu et al., 1991*). The role of CypA as a viral cofactor is best understood for human immunodeficiency virus (HIV-1), where CypA binds to the viral capsid (*Luban et al., 1993*; *Thali et al., 1994*) to regulate interactions with downstream cofactors and protect the capsid and encapsidated viral genome from cellular innate immune sensors (*Rasaiyaah et al., 2013*; *Schaller et al., 2011*; *Kim et al., 2019*). However, the mechanisms by which CypA contributes to other viral infections are less well understood.

Cyps have been implicated in the regulation of viral innate immune evasion (*Rasaiyaah et al., 2013*) and innate immune signalling (*Sun et al., 2014*; *Liu et al., 2017*; *Obata et al., 2005*). In the case of HCV, clinical trials demonstrated that pharmacological inhibition of CypA suppressed HCV

replication and led to elevated type one interferon (IFN) in patients (*Hopkins et al., 2012*). Given the links between CypA and HCV innate immune evasion, we sought to understand the potential roles of CypA in viral innate immune evasion using HCV as a model. Both CypA binding and resistance to cyclophilin inhibitors (CypI) map to the HCV NS5A protein (*Hanoulle et al., 2009*; *Yang et al., 2010*), which has essential roles in HCV replication and assembly (*Ross-Thriepland and Harris, 2015*) and crucially also contributes to immune evasion by several key mechanisms. For example, NS5A is necessary for formation of the membranous replication organelle (RO) (*Romero-Brey et al., 2012*) that cloaks viral RNA replication from cytosolic pattern recognition receptors (*Neufeldt et al., 2016*), preventing innate immune activation. Notably, CypA plays a role in the formation of the RO (*Madan et al., 2014*; *Chatterji et al., 2015*). NS5A also inhibits activation of the key antiviral effector protein kinase R (PKR) (*Gale et al., 1997*) and subsequent PKR-dependent activation of interferon regulatory factor-1 (IRF1)-driven antiviral responses (*Pflugheber et al., 2002*).

Here we have used a panel of novel CypI alongside genetics approaches to discover that CypA regulates HCV evasion of PKR and IRF1 antiviral responses, and that diverse CypI overcome this evasion strategy leading to suppression of virus replication. Our findings advance understanding of CypA-HCV interactions and PKR mechanisms, and open perspectives for the development of novel CypA-targeted therapies that harness host intrinsic antiviral responses to combat infection.

## Results

### CypA is critical for HCV replication in Huh7 cells, but not in Huh7.5 cells

To characterise the role of CypA in HCV innate immune evasion, we took advantage of the human hepatoma cell line Huh7 and its derivative Huh7.5. Huh7.5 cells were selected for enhanced ability to support HCV replication (*Blight et al., 2002*) and spread (*Koutsoudakis et al., 2007*), and also have defective innate immunity (*Sumpter et al., 2005*). We silenced CypA and CypB expression in Huh7 and Huh7.5 cells by stably expressing specific shRNAs (*Figure 1A–B*) and subsequently evaluated HCV replication using the subgenomic replicon (SGR) model. Silencing of CypB expression inhibited HCV replication by ~100 fold in both cell lines (*Figure 1C*), consistent with its previously described role in viral RNA replication (*Watashi et al., 2005*). Intriguingly, silencing of CypA abrogated HCV replication in Huh7 cells but had minimal effect in Huh7.5 cells (*Figure 1C*). We observed the same inhibition profile when the data were normalised to the input luciferase signal at 4 hr post-electroporation (hpe) (*Figure 1—figure supplement 1*), confirming that this observation was not due to differences in electroporation efficiency between cell lines. The differential effect of CypA depletion was mirrored by treatment with the classical CypI cyclosporine A (CsA). CsA abrogated HCV replication in Huh7 cells, but only partially inhibited replication in Huh7.5 cells (*Figure 1D*). CsA also inhibited HCV replication in Huh7.5 cells silenced for CypA expression (*Figure 1E*), with similar antiviral potency regardless of CypA expression (*Figure 1—figure supplement 2*). This suggests that the observed inhibition in Huh7.5 cells may result from inhibition of CypB, which is also a CsA target (*Davis et al., 2010*). This observation explains the controversy of whether CypA or CypB are HCV cofactors, by demonstrating that both have roles, and suggests differential cofactor requirements between Huh7 and more permissive Huh7.5 cells.

We validated the differential cyclophilin dependence in Huh7 and Huh7.5 cells using the HCVcc (J6/JFH1-Rluc) infection model. Silencing of CypA expression, or addition of CsA, completely inhibited HCVcc infection in Huh7 cells, but only weakly inhibited infection in Huh7.5 cells (*Figure 1F–G*), which may reflect a role for CypA in HCV assembly (*Nag et al., 2012*). To further probe this differential requirement for CypA, we evaluated replication of the CypA-independent NS5A D316E/Y317N (DEYN) mutant (*Yang et al., 2010*), which was selected by CypI treatment in highly permissive HCV replicating cell lines. HCV DEYN replicated at wild type levels in Huh7.5 cells, but exhibited a 5-fold replication defect in innate immune competent Huh7 cells (*Figure 1H*). The replication defect in Huh7 cells was partially rescued by addition of CsA (*Figure 1I*). This observation mirrors similar observations for HIV-1 mutants selected to replicate in the presence of CsA in that they too become somewhat dependent on CsA for maximal replication (*Ylinen et al., 2009*). Together, these results confirm that CypA is crucial for HCV replication in innate sensing competent Huh7 cells, but significantly less important for replication in more permissive Huh7.5 cells.

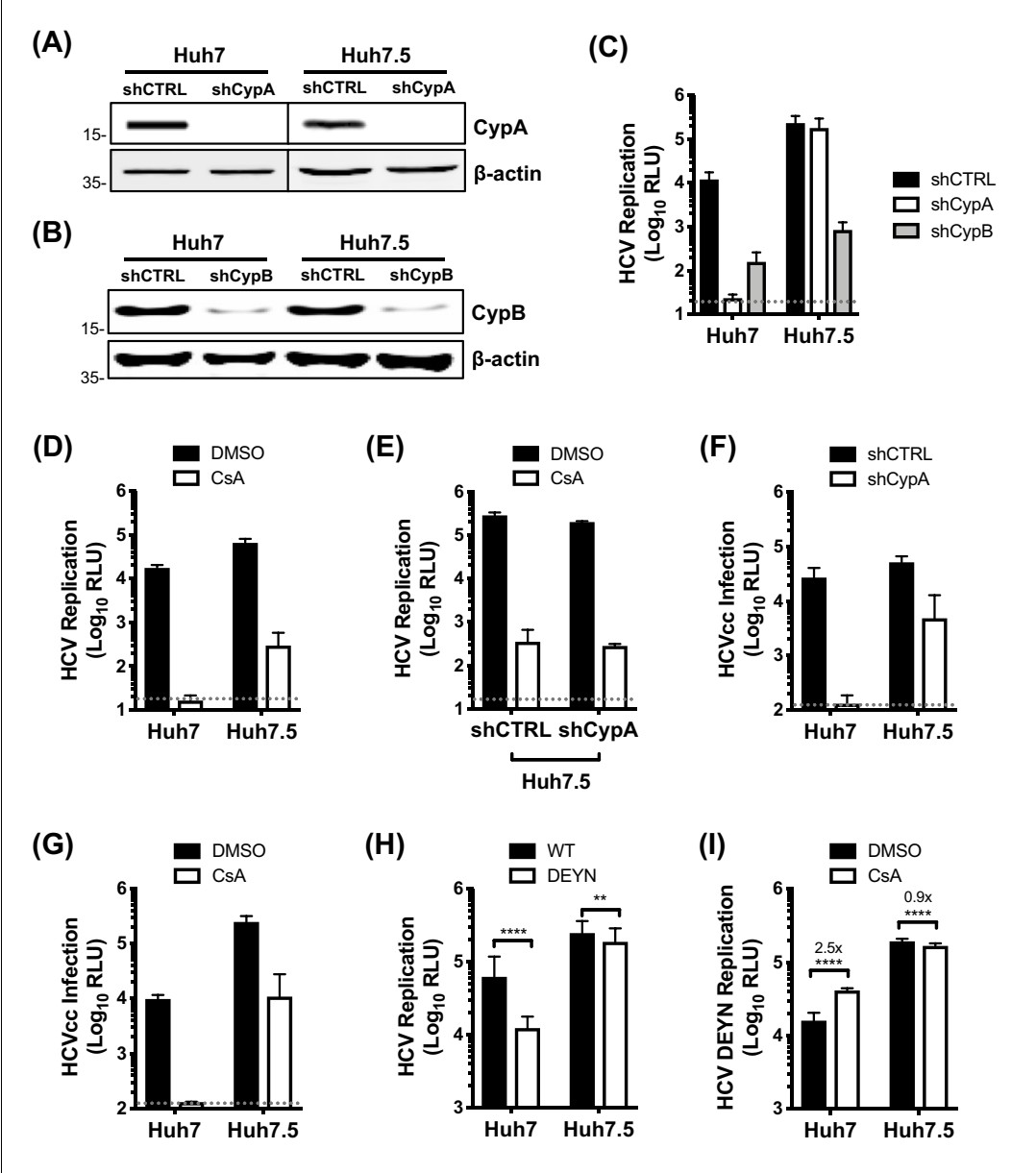

**Figure 1.** CypA is critical for HCV replication in Huh7 cells, but not in Huh7.5 cells. (A) Western blot detecting CypA (A) or CypB (B) expression in Huh7 and Huh7.5 cells transduced with Cyp specific shRNA-expressing lentiviruses as shown. Actin was detected as a loading control. (C) Evaluation of HCV replicon replication in CypA- and CypB-silenced cells. Luciferase reporter activity was measured at 4 and 48 hr post-electroporation (hpe) and is expressed as relative luciferase units (RLU) at 48 hpe. (D-E) Huh7 or Huh7.5 cells (silenced or not for CypA expression) were electroporated with replicon RNA and treated with 1 µM CsA at 4 hr post electroporation (hpe). Luciferase reporter activity was measured at 48 hr post-electroporation. (F) HCVcc infection in CypA-silenced Huh7 and Huh7.5 cells. Cells were infected with HCVcc (J6/JFH1-RLuc) and infection was assessed after 72 hr by measuring luciferase activity. (G) Huh7 or Huh7.5 cells were infected with HCVcc and treated with 1 µM CsA at 4 hr post-infection (hpi). After 72 hr, infection was measured by luciferase activity. (H) Replication of HCV NS5A wild-type (WT) and HCV CsA resistance mutant (NS5A D316E/Y317N; DEYN) in Huh7 and Huh7.5 cells. Cells were electroporated with in vitro transcribed replicon RNA as described above, and replication was assessed by luciferase activity at 48 hpe. (I) Huh7 or Huh7.5 cells were electroporated with HCV NS5A DEYN replicon RNA. After 4 hr, cells were treated with 1 µM CsA and replication was assessed by luciferase activity at 48 hpe. All graphs show relative luciferase units (RLU) expressed as means ± standard deviation from at least three independent experiments each performed in triplicate. Statistical significance was evaluated by t-test using GraphPad Prism (**** p-value<0.0001; ** p-value<0.01). Detection limits of the assays are shown by the dotted grey line.

The online version of this article includes the following figure supplement(s) for figure 1:

**Figure supplement 1.** CypA depletion abrogates HCV replication in Huh7, but not Huh7.5 cells.

**Figure supplement 2.** CsA is equally potent against HCV replication in CypA-depleted Huh7.5 cells.

## Structurally distinct CypI are more potent against HCV replication in Huh7 cells than in Huh7.5 cells

CsA binds to the cyclophilin active site and forms a ternary inhibitory complex with phosphatase calcineurin, which inhibits T cell proliferation leading to immunosuppression (*Liu et al., 1991*). To confirm that the phenotype we observed with CsA treatment (*Figure 1*) was due to cyclophilin binding, and not off-target complex formation, and to generate chemical probes suitable for further mechanistic analyses, we synthesized a panel of novel CypI with distinct structures acting by distinct mechanisms. These included novel CsA analogues and synthetic small molecules based on sanglifehrin chemistry termed depsins (*Figure 2A*). The synthesis and favourable pharmacokinetic properties of depsin molecules were recently described (*Mackman et al., 2018*). Importantly, depsin molecules do not cause immunosuppression via the calcineurin pathway (*Zenke et al., 2001*). We also designed and synthesized a CsA-derived proteolysis targeting chimera (PROTAC; CsA-Prtc1) (*Figure 2A*), which is expected to recruit the von Hippel-Lindau E3 ubiquitin ligase to CsA targets, leading to their proteasomal degradation.

In total, we evaluated ~80 novel CypI for their inhibitory effects on HCV replication in Huh7 cells. We selected the most potent antiviral, and least cytotoxic, molecule from each CypI type for further characterisation (*Figure 2—figure supplement 1*). Using a fluorescence polarisation assay, in which fluorescent CsA is competed from purified recombinant CypA with unlabelled CypI (*Warne et al., 2016*), we first confirmed that the selected molecules bound to CypA with similar nanomolar affinities (*Figure 2B*). CsA-Prtc1 treatment led to degradation of CypA in Huh7 and Huh7.5 cells (*Figure 2C*) within a matter of hours (*Figure 2D*, *Figure 2—figure supplement 2A*) in a dose-dependent manner (*Figure 2E*, *Figure 2—figure supplement 2B*), with complete loss of detectable CypA protein expression occurring within 24 hr at concentrations as low as 100 nM. Degradation was proteasome-dependent (*Figure 2F*) with selectivity for CypA (*Figure 2G–H*), with minimal impact on CypB or CypD protein expression after 24 hr in Huh7 cells, although CypB degradation was observed after 48 hr incubation (*Figure 2—figure supplement 2C*). Moreover, CsA treatment protected CypA from CsA-Prtc1-mediated degradation (*Figure 2I*), illustrating CypA recruitment of CsA within cells.

We next compared the effect of our selected CypI on HCV replication and infection in Huh7 and Huh7.5 cells. Like CsA, our three novel and structurally distinct CypI abrogated HCV replication (*Figure 3A*) and HCVcc infection (*Figure 3B*) in Huh7 cells, but only partially inhibited replication and infection in Huh7.5 cells at the tested dose (*Figure 3A–B*). The CsA-Prtc1 was extremely potent and completely inhibited HCVcc infection in both cell lines at the single dose tested (*Figure 3B*). However, dose-response analyses showed that the CypI were similarly 5- to 10-fold more potent against HCV replication in Huh7 cells compared to Huh7.5 cells (*Figure 3C*), with low nanomolar $IC_{50}$ in Huh7 cells (*Table 1*). The selected CypI were similarly more potent against HCVcc infection of Huh7 cells compared to Huh7.5 cells (*Figure 3D*, *Table 1*). Concordantly, while replication of the NS5A DEYN mutant was resistant to CypI treatment in Huh7.5 cells, CypI treatment dose-dependently enhanced DEYN replication in Huh7 cells (*Figure 3—figure supplement 1*), again illustrating the differential requirements for CypA in the two cell lines during HCV replication.

## Induction of cellular antiviral responses in Huh7 cells, but not in Huh7.5 cells, contributes to the antiviral potency of CypI against HCV

Given that all of the CypI inhibited HCV more potently in innate immune competent Huh7 cells, we hypothesized that the increased potency might be due to induction of effective innate antiviral responses in parental Huh7 that were lacking in the more permissive Huh7.5 cells. To test this, we evaluated expression of IFN-β mRNA by qPCR after treating HCV-replicating or HCVcc-infected Huh7 or Huh7.5 cells with CsA. Consistent with our hypothesis, we observed induction of IFN-β expression in Huh7 cells but not in more permissive Huh7.5 cells (*Figure 4A–B*). Intriguingly, CsA-like molecules and structurally unrelated depsins, but not CsA-Prtc1 (which degrades CypA rather than simply binding CypA and affecting complex formation), were capable of inducing IFN-β expression in HCV-replicating Huh7 cells (*Figure 4C*). One possibility is that activation of IFN-β expression requires the presence of CypA, which is consistent with previous studies suggesting CypA is a necessary co-factor for innate immune sensors, including RIG-I (*Liu et al., 2017*) and PKR (*Daito et al., 2014*).

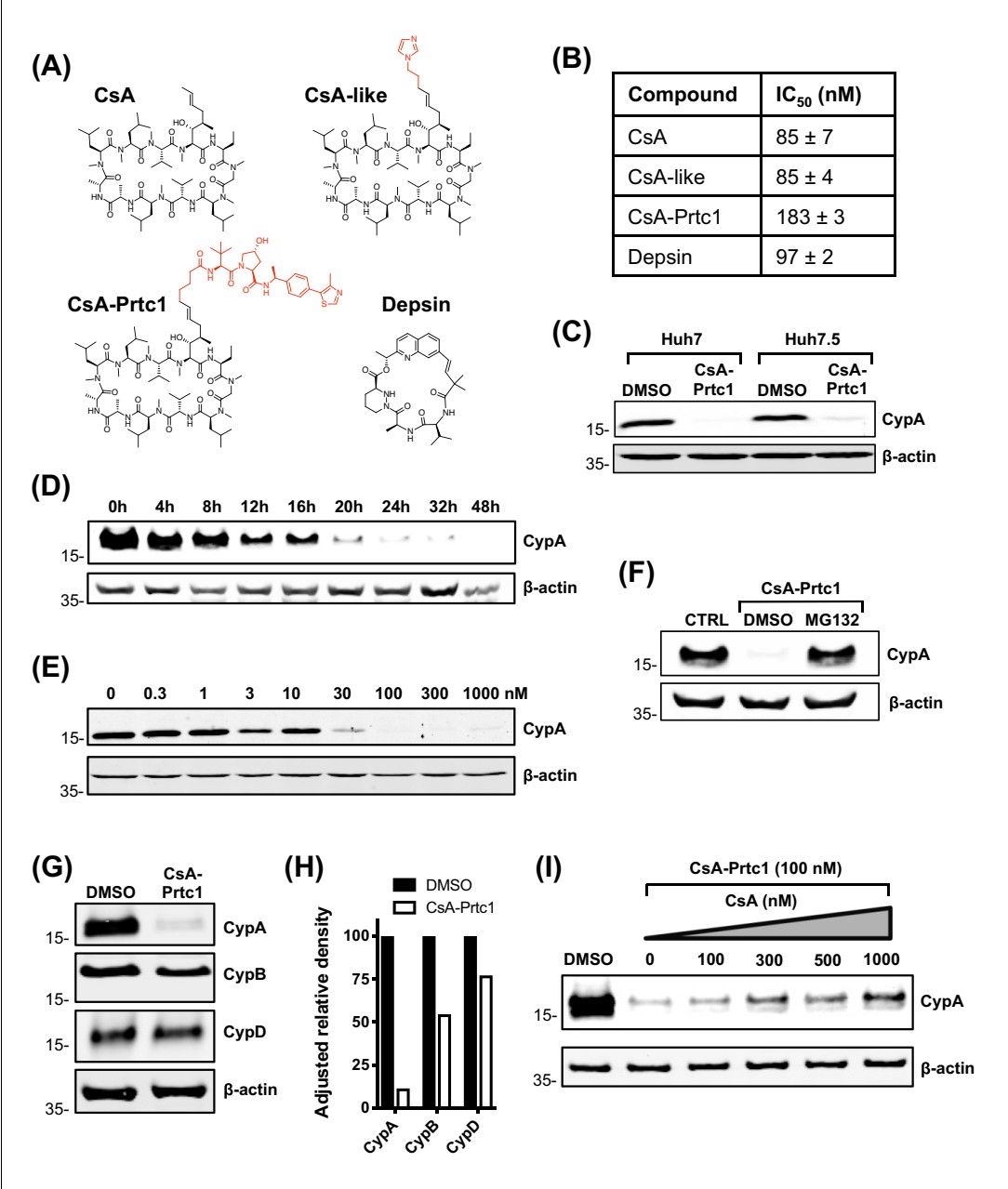

**Figure 2.** Structures and properties of distinct novel CypI. (A) Structures of CypI used in this study; their effects on viral replication and cell viability are shown in Figure supplement 1. (B) CypI-CypA binding affinity measured by fluorescence polarisation using a fluorescein labelled CsA probe. (C) Western blot showing CsA-Prtc1-mediated degradation of CypA in Huh7 and Huh7.5 cells after 48 hr treatment with 1 μM CsA-Prtc1. (D) Analysis of CypA degradation at time points shown after 1 μM CsA-Prtc1 treatment in Huh7 cells detecting CypA expression by western blot. (E) Dose-response of CsA-Prtc1-mediated CypA degradation in Huh7 cells. Cells were treated with the indicated concentrations of CsA-Prtc1 for 48 hr, and CypA levels detected by western blot. (F) CsA-Prtc1-mediated degradation of CypA is proteasome-dependent. Cells were treated with CsA-Prtc1 (1 μM) with or without the proteasome inhibitor MG132 (10 μM) for 24 hr. CypA was detected by western blot. (G) CsA-Prtc1 specificity for CypA. Huh7 cells were treated with 1 μM CsA-Prtc1 for 24 hr and CypA, CypB or CypD detected by western blot. (H) Quantitation by densitometry of gel in (G) showing adjusted relative density normalised to the actin loading control. (I) CsA treatment rescues CypA from CsA-Prtc1-mediated degradation. Huh7 cells, treated for 24 hr with CsA-Prtc1 (100 nM), in the presence of increasing concentrations of CsA, were lysed and CypA expression detected by western blot. (C-I) One representative western blot is shown from at least two independent experiments; (D-E) quantitation by densitometry analysis (showing the combined data from the independent experiments) is shown in Figure supplement 2A-B.

The online version of this article includes the following figure supplement(s) for figure 2:

**Figure supplement 1.** Novel CypI inhibit HCV replication and are not cytotoxic.

**Figure supplement 2.** Characterisation of CsA-Prtc1 activity.

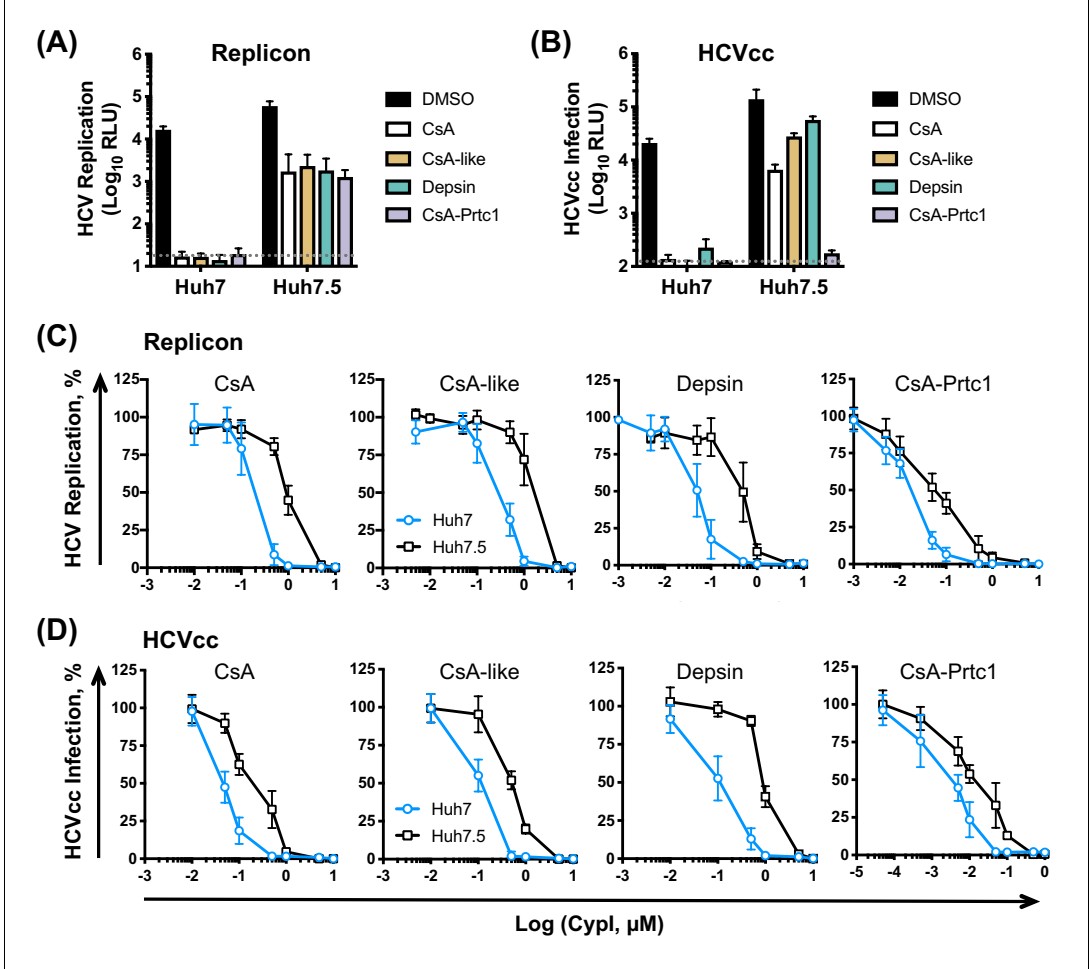

**Figure 3.** CypI are more potent against HCV replication and infection in Huh7 cells than in Huh7.5 cells. (A) CypI more potently inhibit HCV replication in Huh7 cells than in Huh7.5 cells. Huh7 or Huh7.5 cells electroporated with HCV replicon RNA were treated with 1 μM CypI at four hpe and replication was measured by luciferase activity after 48 hr. (B) CypI more potently inhibit HCVcc infection in Huh7 cells than in Huh7.5 cells. Cells infected with HCVcc were treated with 1 μM CypI at four hpi and replication was measured by luciferase activity after 72 hr. (C-D) Dose-response analyses comparing antiviral activity of CypI in Huh7 and Huh7.5 cells. Cells were electroporated with HCV replicon RNA (C) or infected with HCVcc (D) and treated with increasing concentrations of CypI four later. Replication or infection was measured by luciferase activity after 48 hr (C) or 72 hr (D), and is expressed as a percentage relative to the DMSO vehicle-treated control. All graphs show means ± standard deviation from at least three independent experiments each performed in triplicate. Detection limits of the assays are shown by the dotted grey line.

The online version of this article includes the following figure supplement(s) for figure 3:

**Figure supplement 1.** DEYN replication is enhanced by CypI treatment in Huh7 cells.

We confirmed that induction of antiviral responses by CypI in Huh7 cells was specifically the result of HCV replication (rather than transfection of RNA into the cytoplasm) by comparison to a replication-defective replicon with a mutation in the polymerase active site (*Schaller et al., 2007*). Electroporation with either wild-type or replication-defective replicon RNA similarly resulted in translation of luciferase from the input RNA (*Figure 4—figure supplement 1A*) and induction of IFN-β expression at 4 hr post electroporation (hpe) (*Figure 4—figure supplement 1B*), reflecting initial transfection of RNA into the cytoplasm. By 24–48 hpe, however, no luciferase activity or IFN-β mRNA expression above the background level could be detected in cells electroporated with the replication-defective replicon (*Figure 4—figure supplement 1A–B*). Furthermore, the induction of IFN-β expression by CsA at 48 and 72 hpe was only observed in cells electroporated with the wild-type replicon (*Figure 4—figure supplement 1C–F*).

We next evaluated whether the CypI antiviral potency was dependent on increased IFN-β signalling. Addition of exogenous IFN-β inhibited HCV replication in Huh7 cells (*Figure 4D*), confirming

**Table 1.** Comparison of CypI $IC_{50}$ against HCV replication or infection in different cell lines.

| Replication (SGR) | $IC_{50}$ (µM) | | | |
|---|---|---|---|---|
| Cell line | CsA | CsA-like | Depsin | CsA-Prtc1 |
| Huh7 | 0.188 ± 0.032 | 0.266 ± 0.037 | 0.043 ± 0.005 | 0.016 ± 0.002 |
| Huh7.5 | 0.955 ± 0.123 | 1.374 ± 0.197 | 0.336 ± 0.053 | 0.055 ± 0.007 |
| Huh7.5-CTRL | 0.373 ± 0.085 | 0.527 ± 0.154 | 0.206 ± 0.043 | 0.070 ± 0.013 |
| Huh7.5-RIG-I | 0.560 ± 0.105 | 0.686 ± 0.181 | 0.243 ± 0.054 | 0.093 ± 0.019 |
| Huh7.5-Mda5 | 0.754 ± 0.160 | 0.950 ± 0.247 | 0.302 ± 0.060 | 0.100 ± 0.018 |
| Huh7.5-RIG-I/Mda5 | 0.820 ± 0.206 | 1.234 ± 0.298 | 0.351 ± 0.085 | 0.101 ± 0.021 |
| Huh7 NT c7 | 0.068 ± 0.008 | 0.184 ± 0.026 | 0.037 ± 0.004 | 0.009 ± 0.001 |
| Huh7 MAVS KO | 0.112 ± 0.019 | 0.160 ± 0.023 | 0.056 ± 0.008 | 0.010 ± 0.001 |
| Huh7 MAVS KO + C508R | 0.075 ± 0.014 | 0.141 ± 0.028 | 0.045 ± 0.006 | 0.008 ± 0.001 |
| Huh7 PKR KO c1 | 0.176 ± 0.034 | 0.971 ± 0.186 | 0.125 ± 0.021 | 0.021 ± 0.003 |
| Huh7 PKR KO c4 | 0.172 ± 0.040 | 1.249 ± 0.327 | 0.091 ± 0.015 | 0.028 ± 0.004 |
| Huh7 IRF1 KO c10 | 0.183 ± 0.042 | 1.219 ± 0.224 | 0.170 ± 0.039 | 0.027 ± 0.005 |
| Huh7 IRF1 KO c11 | 0.193 ± 0.047 | 1.593 ± 0.284 | 0.192 ± 0.045 | 0.040 ± 0.006 |
| Huh7 + DMSO | 0.216 ± 0.050 | | | |
| Huh7 + C16 | 0.553 ± 0.098 | | | |
| Huh7.5 + DMSO | 0.879 ± 0.131 | | | |
| Huh7.5 + C16 | 1.750 ± 0.495 | | | |
| HCVcc infection | $IC_{50}$ (µM) | | | |
| Cell line | CsA | CsA-like | Depsin | CsA-Prtc1 |
| Huh7 | 0.043 ± 0.012 | 0.091 ± 0.020 | 0.096 ± 0.016 | 0.003 ± 0.001 |
| Huh7.5 | 0.182 ± 0.037 | 0.316 ± 0.083 | 1.084 ± 0.373 | 0.014 ± 0.003 |
| Huh7 NT c7 | 0.033 ± 0.005 | 0.123 ± 0.023 | 0.018 ± 0.003 | 0.012 ± 0.002 |
| Huh7 PKR KO c4 | 0.095 ± 0.019 | 0.433 ± 0.101 | 0.051 ± 0.011 | 0.023 ± 0.005 |

that Huh7 cells are capable of responding to IFN. Although pharmacological inhibition of the Jak/STAT pathway by ruxolitinib treatment rescued HCV replication from inhibition by IFN-β (*Figure 4D*), ruxolitinib treatment had no effect on viral replication in the absence of exogenous IFN-β. Furthermore, ruxolitinib did not affect the potency of CsA (*Figure 4E*) or other CypI (*Figure 4—figure supplement 2*) against HCV replication in Huh7 cells, suggesting that the antiviral effect is independent of IFN signalling. The notion that IFN was not required for maximal CypI activity was also supported by an experiment using anti-human interferon alpha/beta receptor chain two antibody (IFNAR) to inhibit IFN activity through receptor blockade. Anti-IFNAR treatment did not affect the potency of CsA against HCV but effectively rescued HCV replication from inhibition by IFN-β in a control experiment. (*Figure 4—figure supplement 3A–B*). The lack of a requirement for IFN in the effect of CypI is likely explained by direct induction of antiviral genes with inhibitory activity against HCV, here exemplified by viperin (*RSAD2*) (*Helbig et al., 2011*; *Wang et al., 2012*), in Huh7 but not Huh7.5 cells (*Figure 4F*, *Figure 4—figure supplement 4*). Thus, we propose that in these experiments, CypI induces IFN-independent, cell-intrinsic antiviral immunity. However, in vivo, IFN induction would be expected to influence HCV replication and adaptive immune responses and thus the antiviral activity of CypI in patients.

## CypI disrupt formation of the HCV replication organelle

We next sought to identify the mechanisms underlying the observed activation of antiviral genes. CypA has previously been implicated in formation of the HCV RO. Silencing of CypA expression by RNAi (*Chatterji et al., 2015*) or treatment with a CsA analogue (cyclosporine D) (*Madan et al., 2014*) inhibited formation of the double membrane vesicles (DMVs) that comprise the HCV RO, thus inhibiting HCV replication. We hypothesised a model in which 'uncloaking' of viral RNA, aided by

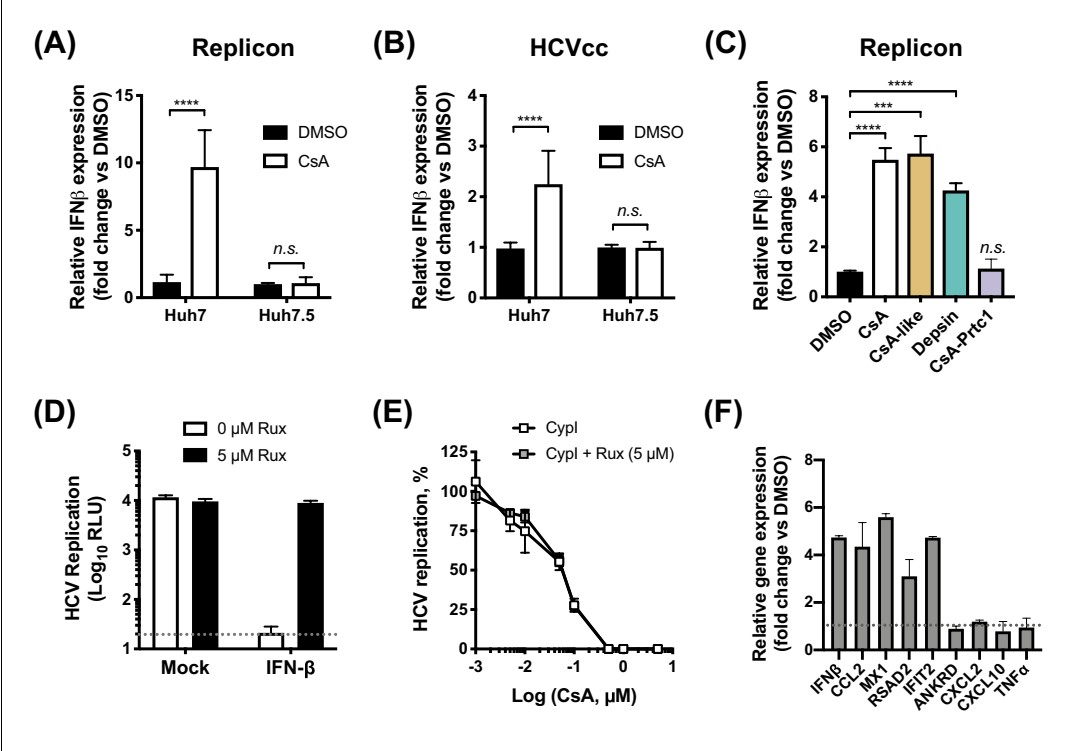

**Figure 4.** CypI induce expression of IFN-β and antiviral genes in Huh7, but not Huh7.5, cells. (A–C) Cells electroporated with HCV replicon RNA (A, C) or infected with HCVcc (B) were treated with 5 µM CsA (A, B) or CypI (C) 4 hr later. After 48 hr, RNA was extracted and expression of IFN-β mRNA was evaluated by qRT-PCR. Data were normalised by GAPDH expression and are expressed as fold change compared to the DMSO vehicle-treated control. (D-E) CypI potency does not depend on IFN signalling. HCV replication in Huh7 cells, electroporated as described above, and treated with IFN-β (5 ng/mL) or CypI, in the presence or absence of the Jak/STAT inhibitor ruxolitinib (Rux). Rux treatment rescued HCV replication from IFN-β inhibition (D) but not from CypI (E). (F) CsA treatment induces expression of a subset of antiviral genes in HCV-replicating Huh7 cells. RNA expression of IFN-β, CCL2, MX1, RSAD2 IFIT2, ANKRD, CXCL2, CXCL10 and TNFα mRNA was evaluated by qRT-PCR at 48 hpe in Huh7 cells electroporated with HCV replicon RNA and treated with CsA (5 µM) at four hpe. Data were normalised by GAPDH expression and are expressed as fold change compared to the DMSO vehicle-treated control. All graphs show means ± standard deviation from at least three independent experiments each performed in triplicate. Statistical significance was evaluated by t-test using GraphPad Prism (**** p-value<0.0001; *** p-value<0.001; n.s. (not significant), p-value>0.05). Detection limits of the assay (D) or gene expression in DMSO-treated cells (set as 1) (F) are shown by the dotted grey line.

The online version of this article includes the following figure supplement(s) for figure 4:

**Figure supplement 1.** Induction of IFN-β expression by CsA depends on HCV replication.

**Figure supplement 2.** Inhibition of IFN-β signalling by ruxolitinib does not affect CypI potency.

**Figure supplement 3.** Inhibition of IFN-β signalling by IFNAR antibody does not affect CsA potency.

**Figure supplement 4.** CsA induces expression of antiviral genes in Huh7, but not Huh7.5, cells.

disruption of the RO, and subsequent sensing of exposed cytosolic viral RNA, leads to IFN production in Huh7 cells. To test this model, we first evaluated whether our novel CypI inhibited formation of HCV-induced DMVs. As described previously, we used an NS3-5B expression construct to specifically evaluate DMV formation independently of viral RNA replication (*Madan et al., 2014*; *Romero-Brey et al., 2012*). Huh7-Lunet/T7 cells were transfected with the pTM-NS3-5B expression construct and treated with CypI 4 hr later. At 24 hr post-transfection, we evaluated NS5A expression and DMV formation. CypI treatment did not affect expression of NS5A as measured by immunofluorescence (*Figure 5A*) and western blot (*Figure 5B*). However, treatment with CsA, depsin or CsA-Prtc1 caused a significant reduction in the number and size of DMVs observed by transmission electron microscopy (TEM) in transfected cells (*Figure 5C–E*), suggesting incomplete and impaired formation of the RO.

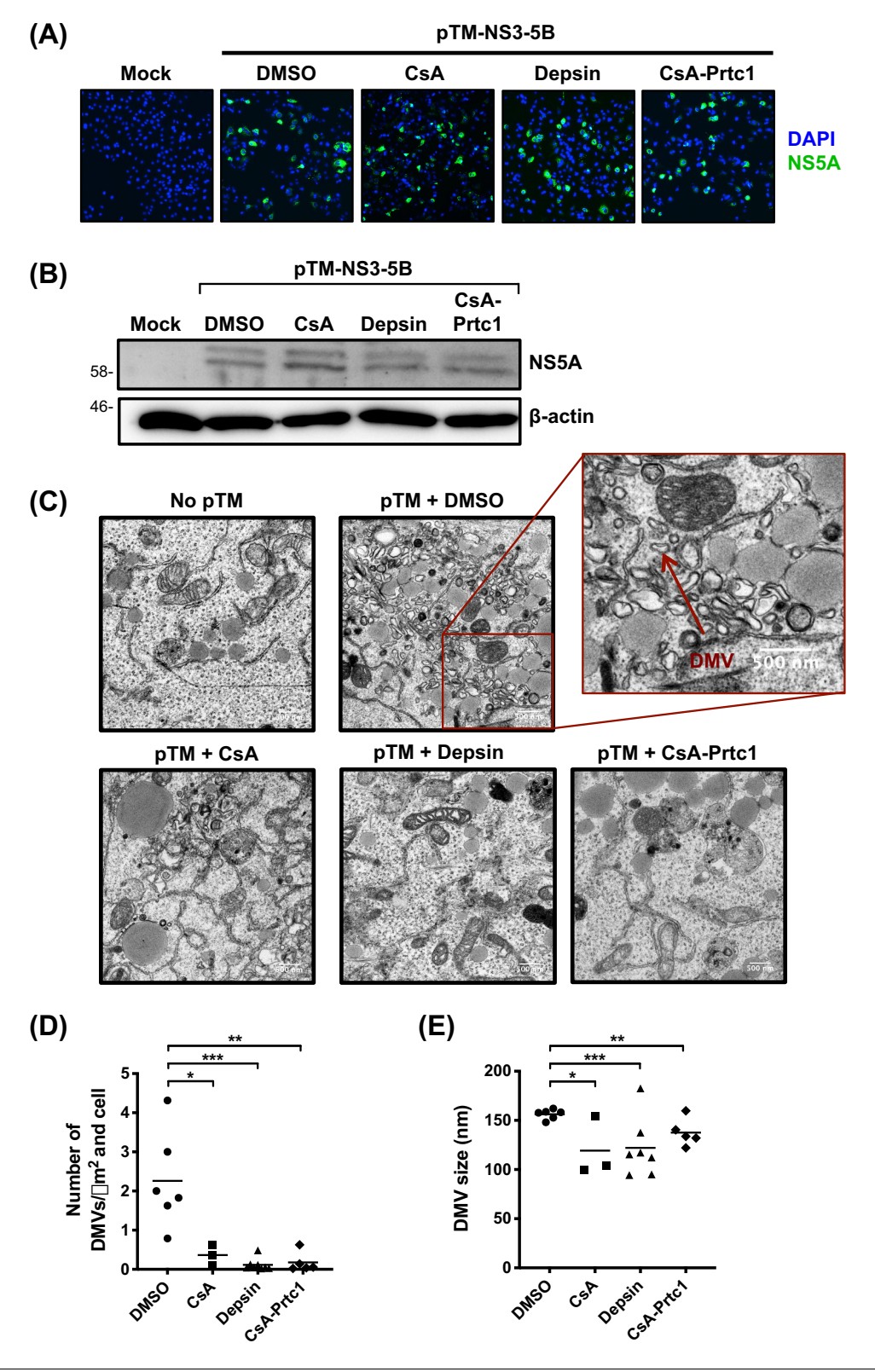

**Figure 5.** Antiviral CypI disrupt formation of the HCV replication organelle. (A-E) Huh7-Lunet/T7 cells were transfected with pTM_NS3-5B and treated with CypI (5X EC90) at 4 hours post-transfection. Transfection efficiency and NS5A expression were evaluated 24 hours later by immunofluorescence (A) or Western blot for NS5A (B). (C) Representative electron micrographs showing the effect of CypI treatment on DMV formation. (D-E) The number and
*Figure 5 continued on next page*

Figure 5 continued

size of DMVs in 3-7 different cells per condition were quantitated using ImageJ. Statistical significance was evaluated by t-test using GraphPad Prism (* p-value < 0.05; ** p-value < 0.01; *** p-value < 0.001).

## RIG-I-like receptors and MAVS do not contribute to the antiviral potency of CypI

Given the proposed role of the RO in viral innate immune evasion, RO disruption may plausibly increase exposure of replicating viral RNA to innate immune sensors. We therefore sought to determine which sensors might contribute to detection of viral RNA in the presence of CypI. Huh7 cells are capable of responding to cytosolic RNA and initiating antiviral signalling through RIG-I, MAVS and IRF3 (*Sumpter et al., 2005*; *Binder et al., 2007*). However, Huh7.5 cells are less responsive, which may reflect a defect in RIG-I (*Sumpter et al., 2005*). Therefore, we hypothesized that the active RIG-I pathway in Huh7 cells contributes to the antiviral signalling induced by CypI. We first evaluated the RIG-I-like receptors (RLRs), RIG-I and Mda5. We tested the activity of CypI against HCV replication in Huh7.5 cells stably expressing functional RIG-I, Mda5, or both RIG-I and Mda5. These reconstituted cell lines have been described previously and have restored RNA sensing of HCV (*Hiet et al., 2015*). However, stable expression of RIG-I or Mda5 (or both) had little, if any, effect on HCV replication at 48 hr (*Figure 6A*) and, more importantly, did not affect the antiviral potency of the CypI in Huh7.5 cells (*Figure 6B*). Similarly, transient transfection of RIG-I into Huh7.5 cells (*Figure 6—figure supplement 1A*) did not affect HCV replication (*Figure 6—figure supplement 1B*), consistent with previous findings (*Binder et al., 2007*), or CypI potency (*Figure 6—figure supplement 1C*).

Since MAVS is a key adaptor protein downstream of RIG-I and Mda5, we confirmed this observation in a loss-of-function context in Huh7 cells by generating clonal Huh7 MAVS knockout (KO) cell lines by CRISPR/Cas9 (*Figure 6C*). We evaluated HCV replication and CypI potency in MAVS KO Huh7 cells compared to control cells generated in the same manner with a non-targeting guide RNA. The loss of MAVS did not affect HCV replication evaluated compared to control cells (*Figure 6D*). However, exogenous expression of WT or HCV protease-resistant MAVS mutant C508R (*Li et al., 2005*) into MAVS KO Huh7 cells decreased viral replication (*Figure 6E*), likely due to induction of interferon responses (*Bender et al., 2015*). This is consistent with the importance of MAVS and its cleavage by HCV protease in HCV replication. Crucially, the antiviral potency of CypI was unaffected by the absence of MAVS (*Figure 6F*), and transfection of the C508R protease-resistant MAVS into MAVS KO Huh7 cells also had no effect on CypI activity (*Figure 6F*). Furthermore, CypI still induced expression of IFN-β mRNA in HCV-replicating MAVS KO Huh7 cells (*Figure 6G*), indicating that CypI induction of IFN-β expression is not dependant on the RLR/MAVS pathway. It is worth noting that treatment with daclatasvir, an inhibitor that similarly blocks formation of the HCV RO (*Berger et al., 2014*) and targets domain I of NS5A (*Gao et al., 2010*), inhibits HCV replication without inducing IFN-β expression (*Figure 6—figure supplement 2A–B*). Therefore, sensing of viral RNA in the presence of CypI is likely not the result of simply 'uncloaking' by disruption of the RO but rather through a more complex CypA-dependent mechanism. Notably, NS5A domain II (where CypA binds) is dispensable for RO formation (*Romero-Brey et al., 2015*) but is required to suppress IFN (*Hiet et al., 2015*) and control PKR (*Gale et al., 1998*).

## PKR modulates the antiviral potency of CypI against HCV

Given the documented role of the CypA target NS5A in binding and inhibiting PKR (*Gale et al., 1997*) and the proposed role of CypA in regulating PKR activity (*Daito et al., 2014*), we next evaluated a role for PKR in determining CypI potency against HCV. We first compared PKR expression and activation in HCV replicating Huh7 and Huh7.5 cells by western blot (*Figure 7A–B*). Importantly, PKR was more abundant in Huh7 cells, and, strikingly PKR T446 autophosphorylation (a marker of PKR activation) was observed in Huh7 but decreased in Huh7.5 cells (*Figure 7A*). We hypothesised that PKR may be impaired in sensing HCV RNA in Huh7.5 cells relative to the parental Huh7 cells. To test this hypothesis, we generated Huh7 PKR KO cell lines by CRISPR/Cas9 (*Figure 7C*) and evaluated the sensitivity of HCV to CypI inhibition in these cells. The antiviral potency of the CypI against HCV replication and HCVcc infection was markedly decreased (*Figure 7D–E*). These data are

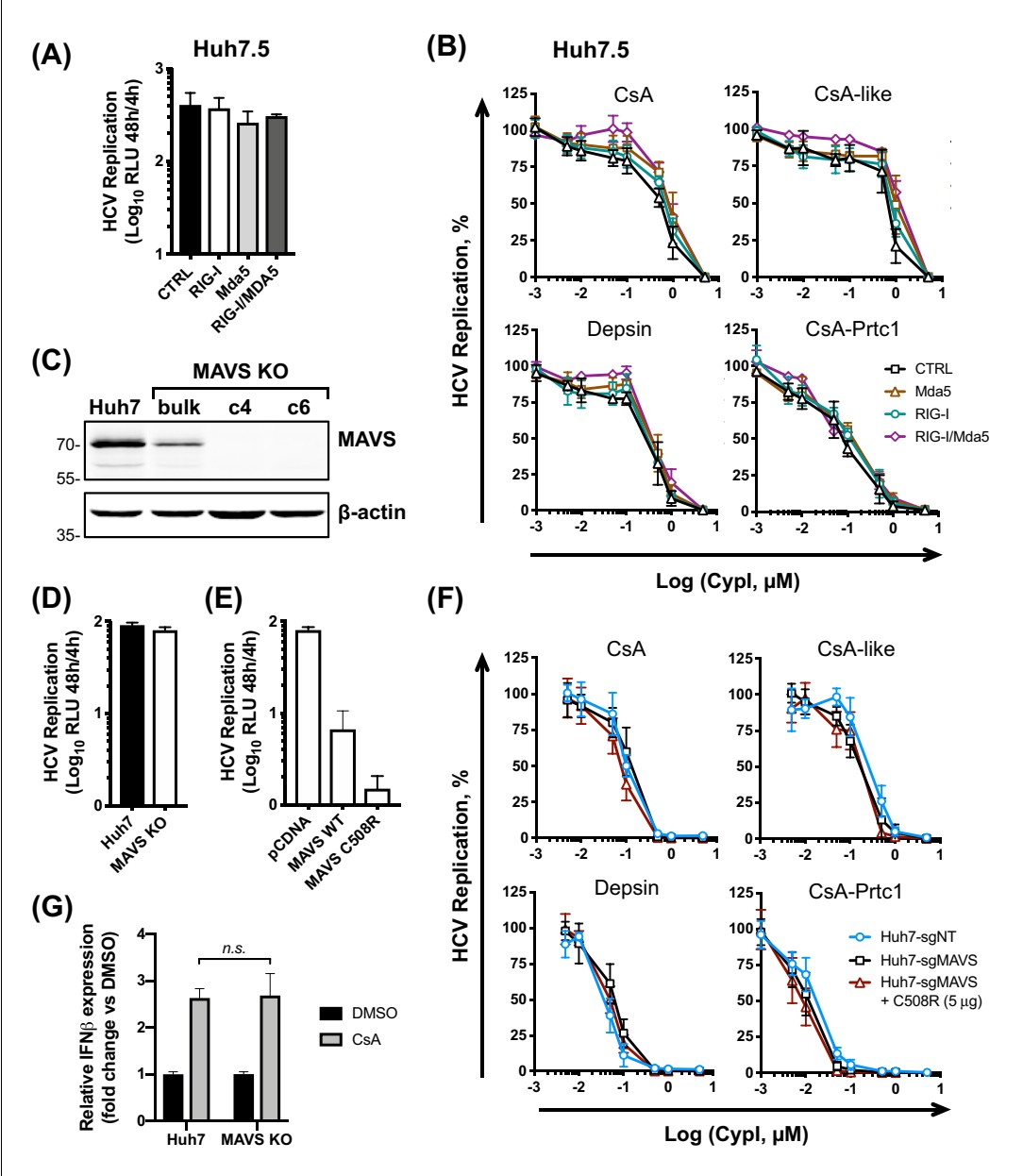

**Figure 6.** The RLR/MAVS pathway does not contribute to the antiviral potency of CypI. Huh7.5 cells stably expressing RIG-I, Mda5 or both were electroporated with HCV replicon RNA and treated with increasing concentrations of CypI at four hpe. Replication was measured by luciferase activity at 48 hpe and is expressed as RLU (A) or percentage relative to the DMSO vehicle-treated control (B). Expression of RIG, Mda5 or both did not significantly affect HCV replication at 48 hr (A) or CypI dose-response curves in Huh7.5 cells (B). (C) Western blot detecting MAVS in single cell cloned Huh7 cells following MAVS knockout by CRISPR/Cas9. Huh7-sgMAVS cells were electroporated with 5 µg HCV replicon RNA in the presence or absence of plasmid encoding wild type MAVS (MAVS-WT) or mutant MAVS-C508R (conferring NS3/4A protease resistance). CypI were added at four hpe. Replication was measured by luciferase activity at 48 hpe and is expressed as RLU (D) or percentage relative to DMSO-treated control (E). HCV replication was not affected by knockout of MAVS (D) but was decreased by transfection of plasmid encoding MAVS-C508R (E). The presence or absence of MAVS did not affect the CypI dose response curves (F). (F) Huh7 or Huh7 MAVS KO cells were electroporated with HCV replicon RNA as described above, and treated with 5 µM CsA at four hpe. At 48 hpe, RNA was extracted and expression of *IFN-β* mRNA was evaluated by qRT-PCR. Data were normalised by GAPDH expression and is expressed as fold change compared to the DMSO vehicle-treated control. (A-F) All graphs show means ± standard deviation from at least three independent experiments each performed in triplicate. Statistical significance was evaluated by t-test using GraphPad Prism (n.s., not significant; p-value>0.05).

The online version of this article includes the following figure supplement(s) for figure 6:

**Figure supplement 1.** Expression of RIG-I in Huh7.5 cells does not affect CypI potency.

**Figure supplement 2.** Daclatasvir treatment does not induce IFN expression in HCV-replicating Huh7 cells.

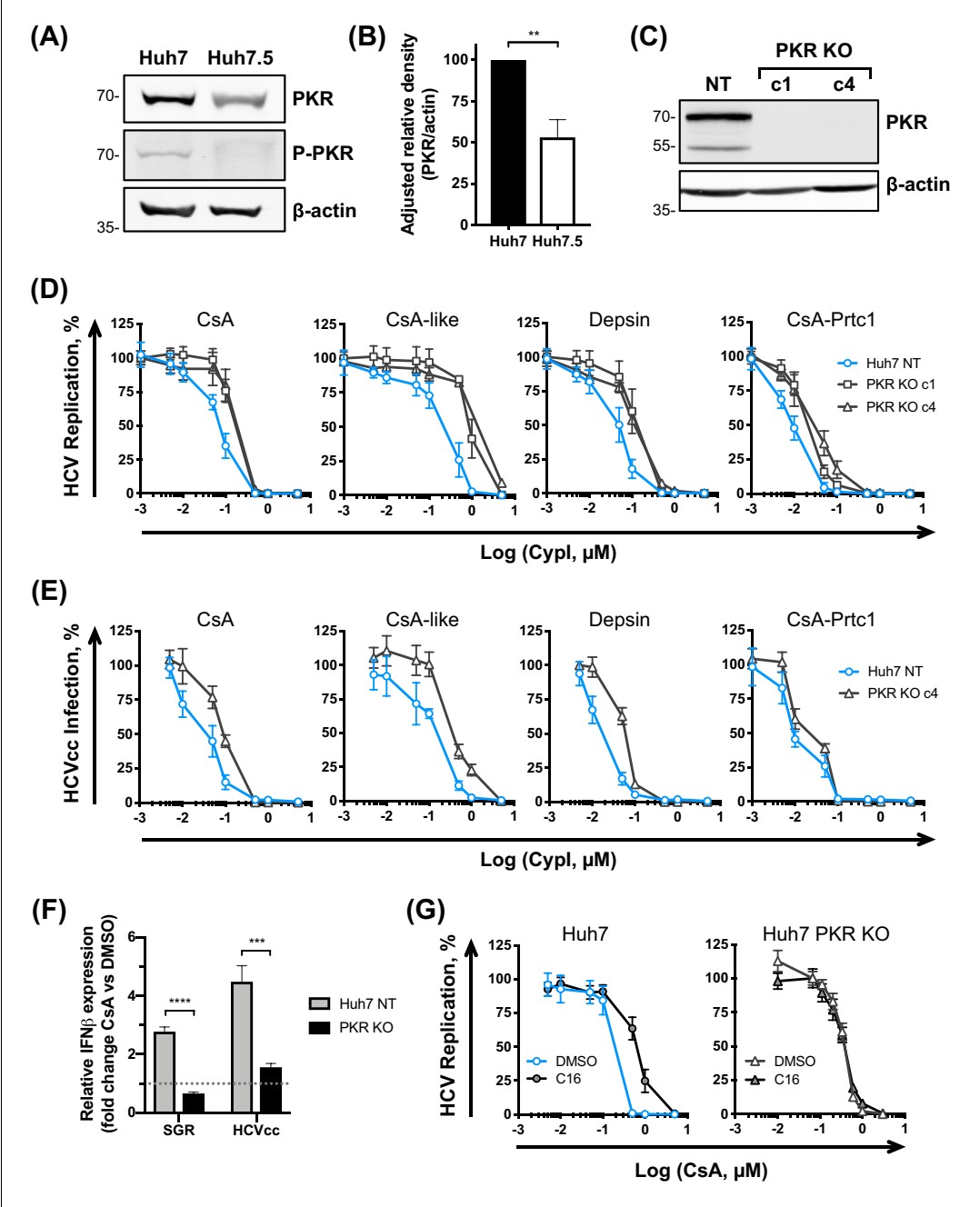

**Figure 7.** PKR modulates the antiviral potency of CypI against HCV. (A) PKR expression and phosphorylation is reduced in Huh7.5 cells. Huh7 or Huh7.5 cells electroporated with in vitro transcribed HCV replicon RNA were lysed at 48 hpe, and PKR expression and phosphorylation assessed by western blot. One representative blot out of three independent experiments is shown. (B) Quantitation of band density from three independent experiments showing adjusted relative density normalised to the actin loading control. (C) Western blot detecting PKR expression in single cell cloned Huh7 cells following PKR knockout by CRISPR/Cas9. (D) CypI potency against HCV replication is decreased in the absence of PKR. Non targeted Huh7 (Huh7 (NT)) or Huh7 PKR KO clones 1 (c1) or 4 (c4) were electroporated with in vitro transcribed HCV replicon RNA and CypI added at four hpe. Replication was measured by luciferase activity at 48 hpe and is expressed as percentage relative to DMSO-treated control. (E) Huh7 NT or PKR KO cells clone 4 (c4) were infected with HCVcc and treated with increasing concentrations of CypI at four hpi. Replication was measured by luciferase activity after 72 hr and is expressed as percentage relative to DMSO-treated control. (F) Huh7 NT or PKR KO cells were electroporated with HCV replicon RNA or infected with HCVcc, and treated with 5 μM CsA at four hpe. At 48 hpe, RNA was extracted and expression of *IFN-β* mRNA was evaluated by qRT-PCR. Data were normalised by GAPDH expression and is expressed as fold change compared to the DMSO vehicle-treated control. (G) Huh7 or Huh7 PKR KO cells were electroporated as described above, and at four hpe were treated with increasing concentrations of CsA in the presence or absence of the PKR inhibitor C16 (1 μM). C16 decreased CypI potency in Huh7 cells, but not in Huh7 PKR KO cells. (A-F) All graphs show means ± standard deviation from

*Figure 7 continued on next page*

*Figure 7 continued*

at least three independent experiments each performed in triplicate. Statistical significance was evaluated by t-test using GraphPad Prism (**** p-value<0.0001; *** p-value<0.001; ** p-value<0.005). Gene expression in DMSO-treated cells (set as 1) (F) shown by the dotted grey line.

The online version of this article includes the following figure supplement(s) for figure 7:

**Figure supplement 1.** PKR inhibitor C16 only minimally affects CsA potency in Huh7.5 cells.
**Figure supplement 2.** PKR overexpression does not affect HCV sensitivity to CsA.
**Figure supplement 3.** PKR does not affect HCV sensitivity to telaprevir or daclatasvir.

consistent with PKR contributing to CypI potency and confirm a role for PKR in the control of HCV replication. Importantly, CsA treatment failed to induce IFN-β mRNA expression in HCV-replicating PKR KO Huh7 cells (*Figure 7F*), further supporting the model in which induction of IFN, and anti-HCV restriction factors, is mediated through PKR in the presence of CypI. To confirm the role of PKR activity, we next tested CypI potency in Huh7 and Huh7.5 cells in the presence of the PKR inhibitor C16, which prevents PKR activation (*Jammi et al., 2003*). C16 decreased CsA potency in Huh7 cells, but not in Huh7 PKR KO cells (*Figure 7G*), and only minimally affected CsA potency against HCV in Huh7.5 cells (*Figure 7—figure supplement 1*).

We also sought to confirm the role of PKR in CypI activity by over-expressing PKR in Huh7 PKR knockout cells. However, PKR over-expression in itself led to PKR activation, as evidenced by its autophosphorylation (*Figure 7—figure supplement 2A*). Concordantly, PKR over-expression had antiviral activity (*Figure 7—figure supplement 2B*) but this did not impact CsA sensitivity (*Figure 7—figure supplement 2C*), presumably because the PKR antiviral effect in this experiment is mediated through activation by phosphorylation and suppression of translation. This is consistent with previous observations of translation shutdown on PKR over-expression (*Grolleau et al., 2000*; *Barber et al., 1993*; *Chong et al., 1992*; *Thomis and Samuel, 1992*).

We next tested whether the absence of PKR broadly affects the sensitivity of HCV to the antiviral activity of telaprevir (NS3/4A protease inhibitor) and daclatasvir (NS5A inhibitor) in Huh7 and Huh7 PKR KO cells. Unlike Cyp inhibition (*Figure 7D–E*), the absence of PKR did not affect the inhibitory activity of telaprevir or daclatasvir (*Figure 7—figure supplement 3*). Collectively, these data are consistent with a specific role for PKR in the enhanced antiviral activity of CypI in Huh7 cells.

## CypI treatment induces PKR- and IRF1-dependent cell-intrinsic antiviral responses

The most well characterised function of PKR is inhibition of RNA translation, which requires PKR autophosphorylation to activate its kinase activity leading to eIF2α phosphorylation. CypI have previously been shown to inhibit PKR autophosphorylation, thus preventing eIF2α phosphorylation by PKR (*Daito et al., 2014*; *Bobardt et al., 2014*), which was proposed to restore expression of ISGs at the protein level to contribute to the antiviral effect (*Daito et al., 2014*). Consistently, we observed that our CypI inhibited PKR autophosphorylation in HCV-replicating cells (*Figure 8—figure supplement 1A*) while only minimally affecting PKR expression based on densitometry analysis (*Figure 8—figure supplement 1B–C*). However, our observation that CypI treatment induces PKR-dependent expression of IFN-β mRNA in HCV-replicating or HCV-infected cells (*Figure 4A–C*, *Figure 7F*) suggests the involvement of additional transcriptional mechanisms.

PKR activates inflammatory transcription factors NF-κB and IRF1 directly and this is thought to be independent of its kinase activity, at least in the case of NF-κB activation (*Bonnet et al., 2000*; *Bonnet et al., 2006*). Interestingly, CsA predominantly induced expression of IRF1 target genes, and not canonical NF-κB targets (*Yamane et al., 2019*; *Figure 4F*), and this induction was PKR-dependent in both HCV-replicating (*Figure 8A*) and HCVcc-infected cells (*Figure 8B*). Thus, restriction of HCV replication by CypI depends on the PKR-mediated induction of antiviral gene expression, including IRF1 targets. Notably, IRF1 and several IRF1 target genes have been shown to negatively regulate HCV replication (*Kanazawa et al., 2004*; *Yamane et al., 2019*), such as *RSAD2* (viperin) (*Helbig et al., 2011*; *Wang et al., 2012*). Unlike CsA, CsA-Prtc1 treatment did not induce expression of IRF1 target genes (*Figure 8—figure supplement 2*), consistent with our earlier findings evaluating IFN-β expression (*Figure 4C*). Importantly, CsA treatment did not induce expression

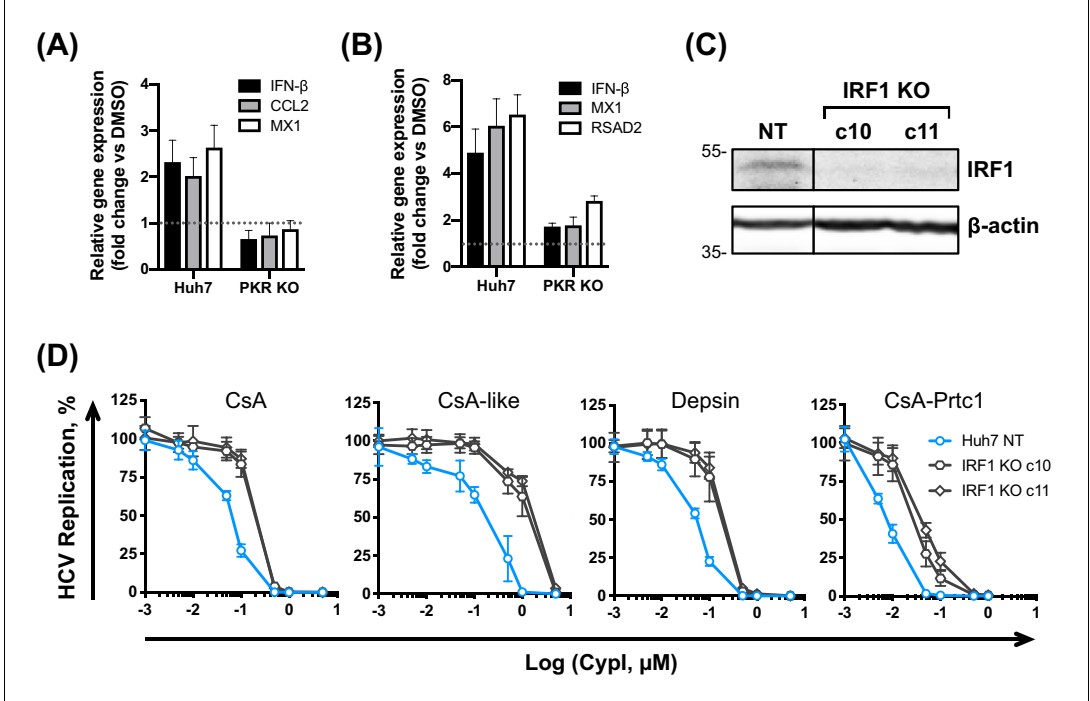

**Figure 8.** PKR induces IRF1-dependent intrinsic antiviral responses in HCV-replicating CypI-treated Huh7 cells. (A–B) Induction of IRF1 target gene expression in HCV-replicating (A) or HCV-infected (B) cells depends on PKR. Expression of *IFN-β, CCL2, MX1* or *RSAD2* mRNA was evaluated by qRT-PCR at 48 hpe in Huh7 NT, or PKR KO cells, electroporated with HCV replicon RNA or infected with HCVcc and treated with 5 µM CsA at four hpe. Data were normalised by GAPDH expression and are expressed as fold change compared to the DMSO vehicle-treated control. (C) Western blot detecting IRF1 in Huh7 cells following IRF1 knockout by CRISPR/Cas9 and single cell cloning. (D) CypI potency against HCV replication was decreased in the absence of IRF1. HCV replication in Huh7 NT or IRF1 KO cells, measured by luciferase activity at 48 hpe, after CypI addition at four hpe, expressed as percentage relative to DMSO-treated control. (A-D) All graphs show means ± standard deviation from two or three independent experiments each performed in triplicate. Gene expression in DMSO-treated cells (set as 1) (A–B) is shown by the dotted grey line.

The online version of this article includes the following figure supplement(s) for figure 8:

**Figure supplement 1.** CypI treatment inhibits PKR autophosphorylation at T446.
**Figure supplement 2.** CsA, but not CsA-Prtc1, induces expression of antiviral genes.

of these genes in the Huh7.5 cells, which we propose are defective for PKR function (*Figure 4—figure supplement 4*).

To confirm the involvement of IRF1, we generated Huh7 IRF1 KO cell lines by CRISPR/Cas9 (*Figure 8C*) and tested the effect of CypI against HCV replication in these cells. Strikingly, the antiviral potency of the CypI against HCV replication was markedly decreased in the IRF1 KO cells (*Figure 8D*). Importantly, the effect of IRF1 KO closely matched the phenotype we observed in the PKR KO cells (*Figure 7D*). Collectively, these findings demonstrate that PKR signalling through IRF1 determines the potency of CypI against HCV.

## Discussion

CypA is a critical host factor for many viruses, although the mechanisms underlying its remarkably broad usage as a viral host factor remain unclear. One intriguing possibility is that viruses recruit CypA to evade host antiviral responses, evidenced by recent studies implicating CypA in viral innate immune evasion (*Rasaiyaah et al., 2013*) and in regulation of innate immune signalling (*Sun et al., 2014*; *Liu et al., 2017*). Therefore, CypA is an exciting target for broadly-acting antiviral intervention based on disrupting viral evasion and harnessing host intrinsic antiviral responses to combat infection. Given that CypA was linked with HCV immune evasion in patients (*Hopkins et al., 2012*), we sought to understand the potential mechanisms in the context of HCV infection. We found that CypA is crucial for HCV evasion of PKR-dependent, but not RLR/MAVS-dependent, antiviral

responses. Furthermore, our results suggest that a deficiency in PKR-dependent responses, as well as defective RIG-I (*Sumpter et al., 2005*), may contribute to Huh7.5 cell permissivity for HCV replication, which is consistent with previous observations showing that RIG-I does not play a role (*Binder et al., 2007*). Indeed, differences in permissivity between Huh7 and Huh7.5 cells are likely governed by a combination of factors, including RIG-I (*Sumpter et al., 2005*) and CD81 (*Koutsoudakis et al., 2007*), among others. Overall, our findings: (a) clarify the role of Cyps in HCV replication, (b) provide mechanistic insight into PKR activity and regulation, and (c) contribute to understanding the broad exploitation of CypA by viruses, opening perspectives for broadly acting antiviral therapies based on disrupting CypA-mediated viral evasion.

## Clarifying the roles of Cyps in HCV replication

While Cyps are clearly involved in HCV replication, the respective roles of Cyp family members have been disputed (*Watashi et al., 2005*; *Yang et al., 2008*). Although binding of CypB to the HCV RNA-dependent RNA polymerase (NS5B) was suggested to be important for HCV RNA replication (*Watashi et al., 2005*), the interaction of CypA with the intrinsically unstructured domain II of HCV NS5A (*Hanoulle et al., 2009*; *Foster et al., 2011*) was also shown to be required for HCV replication and infection (*Yang et al., 2008*; *Kaul et al., 2009*; *Chatterji et al., 2009*; *Liu et al., 2009*).Furthermore, CypA and CypB both bind to proline residues within the unstructured domain II of NS5A (*Hanoulle et al., 2009*; *Foster et al., 2011*), which also interacts with the HCV polymerase NS5B (*Ngure et al., 2016*). Here, our data support a direct role for CypB in HCV RNA replication (*Watashi et al., 2005*), which is consistent with it being equally required in Huh7 and Huh7.5 cells (*Figure 1C*). In contrast, the requirement for CypA varies according to cell line and appears to be important for evasion of host antiviral responses in innate sensing competent cells (i.e., Huh7) (*Figure 1C*).

We propose a model where CypB forms a complex with NS5A and NS5B to directly regulate HCV RNA replication, while CypA forms a complex with NS5A and PKR, leading to inhibition of PKR-dependent antiviral responses (*Figure 9*). CypI, which target both CypA and CypB (*Davis et al., 2010*), directly inhibit HCV replication in both Huh7 and Huh7.5 cells by targeting CypB, and they disrupt the CypA-NS5A interaction, thus rendering them more efficacious in Huh7 cells because there they relieve NS5A inhibition of PKR and restore PKR-dependent antiviral responses. Notably, a differential requirement for CypA in Huh7-Lunet and Huh7.5 cells has also been observed for replication of genotype 1b (Con1) and genotype 2a (JFH-1) replicons (*Kaul et al., 2009*), suggesting that these mechanisms are consistent across HCV genotypes.

CypA has been proposed to have a role in HCV assembly (*Nag et al., 2012*; *Anderson et al., 2011*), which is likely reflected by our observation that CypA depletion in Huh7.5 cells decreased HCVcc infection (*Figure 1F*), but not replication of the HCV replicon (*Figure 1C*). This is consistent with previous studies showing a large decrease in HCVcc infection in Huh7.5 cells silenced for CypA expression (*Gaska et al., 2019*).

## Mechanistic insight into PKR-dependent antiviral responses

NS5A is key for HCV evasion of innate antiviral responses (*Madan et al., 2014*; *Kumthip et al., 2012*; *Park et al., 2012*), including those dependent on PKR (*Gale et al., 1997*; *Gale et al., 1998*; *Pflugheber et al., 2002*). Our data suggest that CypA is required for HCV evasion of PKR-mediated antiviral responses (*Figure 8A–B*) that involve IRF1 signalling (*Figure 4F*). Upon activation by dsRNA binding or cellular stress (*Williams, 1999*), PKR activates IRF1 (*Kirchhoff et al., 1995*), and IRF1 was recently shown to drive intrinsic hepatocyte resistance to positive-sense RNA viruses (including HCV and other *Flaviviridae*) (*Yamane et al., 2019*). Furthermore, IRF1 exerted the most potent inhibition out of 380 ISGs screened against four positive-sense RNA viruses (including HCV, *Flaviviridae* and *Togaviridae*) (*Schoggins et al., 2011*). Consistent with our observations (*Figure 7B–C*), the antiviral activity of IRF1 was independent of canonical Jak-STAT IFN signalling (*Schoggins et al., 2011*), suggesting that IRF1 controls a unique IFN-independent antiviral program that is key for antiviral defense in hepatocytes (*Yamane et al., 2019*). HCV efficiently controls NF-κB (*Kumthip et al., 2012*; *Park et al., 2012*) and MAVS (*Li et al., 2005*), but appears to be more sensitive to IRF1 restriction (*Yamane et al., 2019*). We propose that the ability of HCV to inhibit PKR and evade

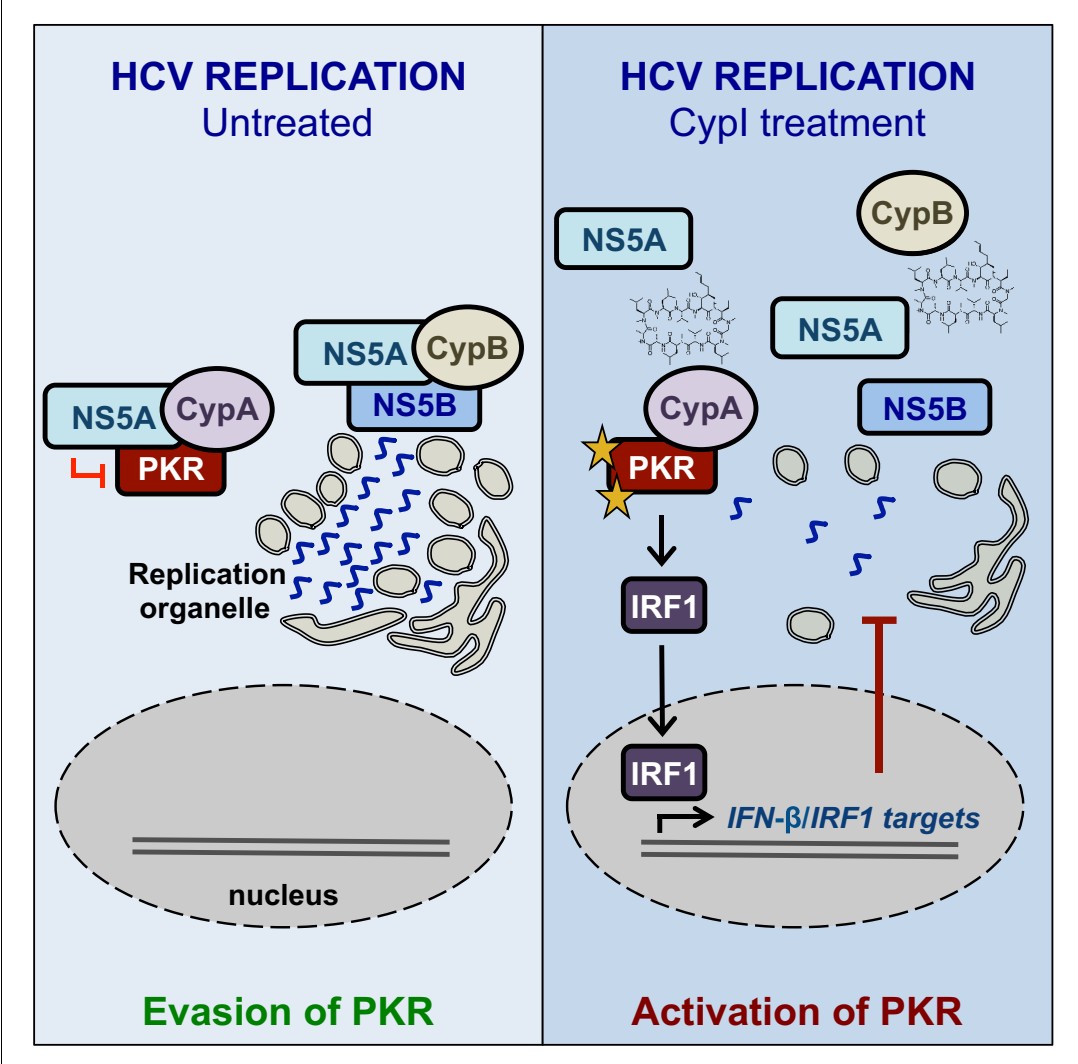

**Figure 9.** Working model for the proposed roles of Cyps in HCV replication, and the proposed antiviral mechanisms of CypI against HCV.

downstream IRF1 responses depends critically on the CypA-NS5A interaction, and that disruption of this interaction by CypI leads to activation of PKR and engages IRF1-dependent antiviral responses.

CypA has been proposed as a regulator of PKR activity in the context of eIF2α phosphorylation (*Daito et al., 2014*). Our data suggest that CypA is required for regulation of PKR in a broader context, including activation of IRF1, since targeting CypA for proteolytic degradation by PROTAC (e.g. CsA-Prtc1) blocked the activation of IFN-β or IRF1 target gene expression that we observed with CsA (*Figure 4C*, *Figure 8—figure supplement 2*). In contrast to CsA-Prtc1, CsA disrupts CypA interactions and therefore affects complex formation, but does not cause CypA degradation. Given that CypA regulates protein complexes and that CypA, NS5A and PKR have all been shown to interact (*Hanoulle et al., 2009*; *Daito et al., 2014*; *Gale et al., 1998*), CypA-NS5A-PKR complex formation likely regulates PKR activation in HCV-replicating cells. Our data suggest that perturbation of the complex by CypI disrupts NS5A inhibition of PKR, but not PKR activation of IRF1.

The regulation of PKR is not fully understood (*Bou-Nader et al., 2019*), and the mechanisms by which PKR activates NF-κB and IRF1 in particular are unclear, although appear to be independent of PKR kinase activity, at least in the case of NF-κB (*Bonnet et al., 2000*; *Bonnet et al., 2006*). Our data suggest that activation of IRF1 by PKR does not require canonical PKR activation mechanisms (i.e. autophosphorylation), as CypI inhibit PKR autophosphorylation in HCV-replicating cells (*Figure 8—figure supplement 1*) but still induce PKR-dependent activation of antiviral target genes

(*Figure 8A–B*) regulated by IRF1 and potentially other transcription factors, suggesting distinct mechanisms of PKR activation and downstream signalling.

## A novel antiviral mechanism

Using HCV as a model, we have discovered that CypI restore PKR-dependent antiviral responses to inhibit infection. This represents a novel and potentially broadly acting antiviral mechanism based on inhibition of viral evasion and restoration of host intrinsic antiviral immunity. Whether other viruses exploit the CypA-PKR interaction to evade antiviral immunity remains to be determined, although many viruses do require CypA (*Dawar et al., 2017*), including medically important (and currently untreatable) human viruses such as *Flaviviridae* and *Coronaviridae* family members (*Qing et al., 2009*; *de Wilde et al., 2011*; *Pfefferle et al., 2011*). Many viruses also encode PKR antagonists, although the roles of CypA in viral PKR antagonism (beyond our current study) have not yet been elucidated.

Overall, CypA is an attractive antiviral target for a broad array of viruses, including emerging human viruses currently lacking specific antiviral therapies. Here, we contribute to the understanding of CypA-HCV interactions and PKR activation, opening perspectives for the further development of CypA-targeting broadly acting antivirals against untreatable human viruses.

# Materials and methods

## Cell lines

Huh7 and Huh7.5 cells were kindly provided by Dr. Joe Grove (UCL). Huh7 cells were originally obtained from Dr. Yoshiharu Matsuura (from Japanese Collection of Research Bioresources Cell Bank, JCRB0403). Huh7.5 cells were obtained from Apath LLC (APC166). Huh7, Huh7.5 and 293 T cells were cultured in DMEM supplemented with 10% FBS, 50 U/mL penicillin and 50 µg/mL streptomycin at 37˚C in 5% $CO_2$. Huh7-Lunet/T7 cells (*Friebe et al., 2005*) were cultured in the presence of 5 µg/mL Zeocin. Cells routinely tested negative for mycoplasma using the Lonza MycoAlert mycoplasma detection kit.

## Inhibitors

Synthesis of the novel CypI is described in *Supplementary file 1*. CypI were resuspended in dimethyl sulfoxide (DMSO, Sigma-Aldrich) as 10 mM stocks. CypI were diluted in DMEM-10% FBS at the indicated concentrations and added to cells at 4 hr post-electroporation unless otherwise indicated. The PKR inhibitor C16 was obtained from Sigma-Aldrich (I9785). Ruxolitinib was obtained from Cell Guidance Systems. Telaprevir (VX-950) was obtained from Generon/Adooq Bioscience (A10902-2). Daclatasvir (BMS-790052) was obtained from Insight Biotechnology (D101505).

## Antibodies

Mouse monoclonal anti-NS5A antibody (9E10) was kindly provided by Dr. Joe Grove (UCL) and has previously been described (*Lindenbach et al., 2005*). Antibodies against β-actin (Abcam; ab8226 or ab8227), CypA (Enzo; BML-SA296-0100), CypB (Abcam; ab16045), CypD (Abcam; ab110324), RIG-I (Cell Signaling Technology; #3743) MAVS (Santa Cruz Biotechnology; sc166583), PKR (Abcam; ab32052) and phospho-PKR T446 (Abcam; ab32036) were also used. Secondary IRDye 680- or 800-labelled antibodies and AlexaFluor-conjugated antibodies were obtained from LI-COR Biosciences or Thermo Scientific, respectively. Anti-human interferon alpha/beta receptor chain two antibody (IFNAR2) (Pbl Assay Science, 21385–1) and an IgG2A control antibody (R and D Systems, 4460 MG-100) were used at 2 µg/mL.

## Plasmids

The subgenomic reporter replicon pFKI389Luc/NS3-3’_dg_JFH (HCV SGR) and replication deficient mutant with a deletion in the NS5B active site (ΔGDD) have been described previously (*Schaller et al., 2007*). The plasmid J6/JFHRluc2 (HCVcc) was kindly provided by Dr. Joe Grove (UCL) with permission from Apath LLC. The HCV polyprotein expression construct pTM_NS3-5B has been described previously (*Romero-Brey et al., 2012*). LentiCRISPRv2 was a gift from Dr. Feng Zhang (Addgene plasmid #52961). For knockout of specific genes, synthetic oligos

(*Supplementary file 2*) were cloned into LentiCRISPRv2 as described (*Sanjana et al., 2014*). The lentiviral PKR expression plasmid (pSCRPSY-EIF2AK) (*Feng et al., 2018*) was a kind gift from Dr. Sam Wilson (University of Glasgow).

## In vitro transcription and electroporation of RNA

Plasmid DNA (10 µg) was linearised by digestion with MluI (HCV SGR) or XbaI (HCVcc). Purified linearised DNA (1 µg) was used as a template for in vitro transcription according to the T7 MEGAscript Kit instructions (Ambion, Life Technologies). RNA was resuspended in nuclease-free water at a concentration of 1 µg/µL, aliquoted and stored at $-80°C$. HCV SGR RNA (5 µg) or HCVcc RNA (10 µg) was electroporated into $2 \times 10^6$ cells or $4 \times 10^6$ target cells, respectively, using either a Neon transfection system (Thermo Scientific) or Amaxa nucleofector (Lonza). In both cases, single-cell suspensions were washed with PBS and resuspended in 100 µL of Buffer R (Neon) or Nucleofector Solution T (Amaxa), respectively. Resuspended cells were mixed with RNA and loaded into a Neon Tip or Amaxa cuvette. Cells were electroporated using the Neon Transfection system (1400 V, 20 ms, one pulse) or the Amaxa Nucleofector system (program T-016) and resuspended in DMEM-10% FBS prior to seeding in 96-well plates at a density of $\sim 2 \times 10^4$ cells/well. For experiments using IFNAR2 antibody, electroporated cells were plated at $\sim 7.5 \times 10^3$ cells/well.

## Preparation of virus stocks

In vitro transcribed RNA (10 µg), generated as described above, was electroporated into $4 \times 10^6$ Huh7.5 cells. Electroporated cells were plated into 6-well plates, and were split and expanded as necessary. Supernatants containing HCVcc were collected on days 3 and 7, and filtered through a 0.45 µm syringe filter.

## Lentivirus production and generation of stable cell lines

HEK293T cells plated in 10 cm dishes were transfected with 1 µg packaging plasmid p8.91 (*Zufferey et al., 1997*), 1 µg envelope plasmid pMDG encoding VSV-G protein (*Naldini et al., 1996*) and 1.5 µg of transfer plasmid pHIV-SIREN (*Schaller et al., 2011*) encoding shRNA or lentiCRISPRv2 (*Sanjana et al., 2014*) encoding sgRNA or pSCRPSY-EIF2AK2 lentiviral plasmid encoding PKR (*Feng et al., 2018*) using Fugene-6 transfection reagent (Promega) as described (*Fletcher et al., 2015*). Lentivirus supernatants were collected at 48 hr and 72 hr post-transfection and clarified by filtration through 0.45 µm syringe filters. Huh7 or Huh7.5 cells were plated in 6-well plates at a density of $2.5 \times 10^5$ cells/well prior to being transduced with 1 mL/well of lentivirus supernatant in the presence of 8 µg/mL polybrene. Transduced cells were selected by addition of 2.5 µg/mL puromycin at 72 hr post-transduction. Alternatively, to generate CRISPR knockout cells without genome integration of Cas9, the lentiCRISPRv2 plasmid (2.5 µg) was electroporated into Huh7 or Huh7.5 cells ($5 \times 10^5$ cells) using the Neon electroporator as described above. Electroporated cells were plated in 10 cm² dishes and selected by addition of 2.5 µg/mL puromycin at 24 hr post-electroporation. After 72 hr of puromycin selection, single cell clones were isolated by limiting dilution in 96-well plates. Loss of target protein expression was confirmed by western blot.

## Site-directed mutagenesis

The HCV NS5A D316E/Y317N mutant (*Yang et al., 2010*) was generated in the subgenomic replicon using a modified version of the Q5 Site-Directed Mutagenesis Protocol (New England BioLabs). The PCR reaction was assembled according to the protocol, using mutagenic primers (for oligos see *Supplementary file 2*). PCR product (4 µL) was used in the subsequent kinase-ligase-Dpn1 reaction, following which 5 µL of ligation product was transformed into chemically competent *E. coli* (strain HB101). The mutation was confirmed by sequencing using an NS5A forward primer (*Supplementary file 2*). The MAVS C508R mutation conferring NS3/4A protease resistance (*Li et al., 2005*) was generated using a modified version of the QuikChange II Site-Directed Mutagenesis (Agilent) protocol. The PCR was assembled using the MAVS-WT plasmid (50 ng) as template with *Pfu* Ultra High Fidelity polymerase (Agilent) and the mutagenic primers (*Supplementary file 2*). The PCR product was incubated with DpnI restriction enzyme (10 U/µL) at 37°C for 1 hr and then transformed into *E.coli* HB101. The mutation was confirmed by sequencing using a CMV forward primer (*Supplementary file 2*).

## Luciferase measurement

Firefly luciferase activity was measured using the SteadyGlo reagent according to the manufacturer instructions (Promega). For measurement of Renilla luciferase activity, cells were washed once with PBS and then lysed with 50 µL/well of 1X passive lysis buffer (Promega). Lysates (20 µL) were transferred to 96-well white plates and Renilla activity was measured following addition of 50 µL of 2 µg/mL coelenterazine (NanoLight).

## Transmission electron microscopy

Huh7-Lunet/T7 cells were seeded onto glass coverslips at a density of $1 \times 10^5$ cells/well. Cells were transfected 24 hr later with the pTM_NS3-5B polyprotein expression construct using the TransIT LT1 transfection reagent (Mirus Bio LLC, Madison, WI). After 4 hr, cells were treated with DMSO or CypI (at 5X $EC_{90}$, corresponding to 5 µM for CsA, 2 µM for depsin and 1 µM for CsA-Prtc1) until fixation 21 hr later. Cells were fixed for 30 min at room temperature with 2.5% glutaraldehyde in 50 mM sodium cacodylate (caco) buffer (pH 7.2) containing 10 mM $MgCl_2$, 10 mM $CaCl_2$, 100 mM KCl and 2% sucrose. Following washes with 50 mM caco buffer, cells were incubated with 2% osmium tetroxide in caco buffer for 40 min on ice. Samples were then washed with distilled water and incubated in 0.5% uranyl acetate overnight at 4°C. Samples were then washed with distilled water, and dehydrated by sequential incubation with increasing concentrations of ethanol (40%, 50%, 60%, 70%, 80%, 95%, 100%). Dehydrated samples were embedded in araldite-Epon (Araldite 502/Embed 812 kit, Electron Microscopy Sciences) and polymerized for 2 days at 60°C. Embedded cells were then cut into 70 nm thin sections (Leica Ultracut UCT microtome) and mounted onto a mesh grid. Sections were contrasted by incubation with 3% uranyl acetate in 70% methanol for 5 min, followed by incubation with 2% lead citrate in distilled water for 2 min. Finally, sections were visualised using a JEOL JEM1400 transmission electron microscope (JEOL Ltd., Tokyo, Japan) in the Electron Microscopy Core Facility at Heidelberg University. Images were analysed and double membrane vesicles were counted using ImageJ.

## Immunofluorescence

Huh7-Lunet/T7 cells seeded onto glass coverslips at a density of $1 \times 10^5$ cells/well were fixed in 4% paraformaldehyde and then washed three times with PBS. Cells were then incubated with NS5A-specific monoclonal antibody (9E10) diluted 1:1000 in PBS containing 1% FBS and 0.5% Triton X-100. After overnight incubation at 4°C, cells were washed and secondary donkey anti-mouse AlexaFluor-488 antibody was added. Nuclear DNA was detected by DAPI staining. After incubation at room temperature for 1 hr, coverslips were washed, mounted on slides with FluoromountG and sealed with clear nail polish and imaged using a Nikon Eclipse Ti with 10x objective.

## qRT-PCR

Cellular RNA was extracted using the RNeasy Mini kit (Qiagen) according to the manufacturer instructions. Recovered RNA was quantitated by Nanodrop and 500 ng of RNA was used to synthesize cDNA following the Superscript III Reverse Transcriptase protocol (Invitrogen). The resulting cDNA was diluted 1:5 in nuclease-free water prior to quantitative PCR (qPCR) using the FastSYBR Green Master Mix (Applied Biosciences). Reactions contained 5 µL 2X FastSYBR Green master mix, 2 µL diluted cDNA, 1 µL forward primer, 1 µL reverse primer and 1 µL nuclease-free water. Expression of IFN-β, ISGs and glyceraldehyde-3-phosphate dehydrogenase (GAPDH) was determined using specific primers (primer details in Table S1). Following normalisation to GAPDH expression, IFN-β or ISG expression was calculated as fold increase relative to DMSO-treated cells.

## Western blot

Cells were resuspended in cell lysis buffer (50 mM Tris pH8, 150 mM NaCl, 1 mM EDTA, 10% glycerol, 1% Triton X-100% and 0.05% NP40). Cell lysates were incubated on ice for 30 min, followed by centrifugation at 14,000 rpm at 4°C for 15 min. Samples were diluted in 4X SDS-PAGE loading buffer (200 mM Tris pH 6.8, 8% SDS, 0.4% bromophenol blue, 40% glycerol and 2% β-mercaptoethanol), heated at 95°C for 5 min, and loaded onto 10% or 15% polyacrylamide-SDS gels. Following electrophoresis, proteins were transferred to a nitrocellulose membrane using the Bio-Rad TransBlot Turbo system according to the manufacturer instructions. Membranes were blocked in 5% milk diluted in

Tris-buffered saline (TBS) with 0.5% Tween (TBS-T) for 1 hr prior to incubation with primary antibodies diluted in blocking solution overnight at 4°C. Membranes were washed extensively in TBS-T prior to incubation for 1 hr at room temperature with IRDye 800-labelled or IRDye 680-labeled antibodies diluted 1:10,000 in blocking solution. Membranes were washed extensively in TBS-T followed by washes in TBS (without Tween) and then scanned using an Odyssey Infrared imaging System (LI-COR Biosciences). Alternatively, membranes were incubated with horseradish peroxidase-conjugated mouse-specific secondary antibodies (Sigma-Aldrich, St. Louis, MO) diluted 1:10,000, prior to detection with the Western Lightning Plus-ECL reagent (Perkin-Elmer, Waltham, MA) and the Intas Science imager. Where indicated, densitometry analyses were performed using ImageJ and expressed as adjusted band density (normalized to actin loading control).

### Cell viability assay

Huh7 or Huh7.5 cells were seeded in 96-well plates at a density of $1 \times 10^4$ cells/well prior to being treated with serially diluted CypI. After 48 hr, cell viability was assessed using the alamarBlue Cell Viability Assay (ThermoScientific) according to the manufacturer instructions. Absorbance was measured using a microplate reader (Multiskan FC Microplate reader, Thermo Scientific) at 570 nM with a reference measurement at 595 nm.

## Acknowledgements

We are grateful to Joe Grove (UCL) for kindly sharing HCV reagents. We thank Rebecca Sumner and Lucy Thorne (Towers lab, UCL) for sharing CRISPR and shRNA reagents, and all members of the Towers laboratory for helpful discussions. We thank Lena Werstein and Uta Hasselman (Bartenschlager lab, Heidelberg University) for invaluable assistance in preparing samples for EM and Richard Milne (UCL) for helpful comments on the manuscript. CCC thanks Christopher Lohans (Queen's University) for helpful discussions and ChemDraw expertise. CCC was supported by fellowships from the Canadian Institutes of Health Research (201411MFE-338606–245517) and the Canadian Network on Hepatitis C, and a Research Initiation Grant from Queen's University. This work was also supported by grants from the National Institute for Health Research University College London Hospitals Biomedical Research Centre (GJT), a Medical Research Council Project Grant (CCC, GJT), a Wellcome Trust Senior Biomedical Research Fellowship (GJT) and the European Research Council under the European Union's Seventh Framework Programme (FP7/2007-2013)/ERC (grant HIVInnate 339223). RB was supported by the Deutsche Forschungsgemeinschaft (DFG, German Research Foundation) – project number 272983813 – TRR 179.

## Additional information

### Funding

| Funder | Grant reference number | Author |
| --- | --- | --- |
| Canadian Institutes of Health Research | 201411MFE-338606-245517 | Che C Colpitts |
| Wellcome | Wellcome Trust Senior Biomedical Research Fellowship 108183 | Greg J Towers |
| National Institute for Health Research | University College London Hospitals Biomedical Research Centre grant | Greg J Towers |
| European Commission | European Research Council 7th Framework Programme (FP7/2007-2013)/ERC (Grant HIVInnate 339223) | Greg J Towers |
| Deutsche Forschungsgemeinschaft | 272983813 - TRR 179 | Ralf Bartenschlager |
| Faculty of Health Sciences, Queen's University | Research Initiation Grant | Che C Colpitts |

CanHepC - Canadian Network                          Che C Colpitts
on Hepatitis C

The funders had no role in study design, data collection and interpretation, or the
decision to submit the work for publication.

## Author contributions

Che C Colpitts, Conceptualization, Resources, Data curation, Formal analysis, Supervision, Funding acquisition, Validation, Investigation, Methodology, Project administration; Sophie Ridewood, Formal analysis, Investigation, Methodology, Writing - review and editing; Bethany Schneiderman, Formal analysis, Validation, Investigation; Justin Warne, Resources, Formal analysis, Investigation; Keisuke Tabata, Resources, Formal analysis, Investigation, Methodology; Caitlin F Ng, Validation, Investigation; Ralf Bartenschlager, Resources, Supervision, Funding acquisition, Methodology; David L Selwood, Resources, Formal analysis, Supervision, Investigation, Methodology; Greg J Towers, Resources, Supervision, Funding acquisition, Project administration

## Author ORCIDs

Che C Colpitts (iD) https://orcid.org/0000-0003-2474-1834
Greg J Towers (iD) https://orcid.org/0000-0002-7707-0264

## Decision letter and Author response

Decision letter https://doi.org/10.7554/eLife.52237.sa1
Author response https://doi.org/10.7554/eLife.52237.sa2

# Additional files

## Supplementary files

- Supplementary file 1. Synthesis of novel CypI.
- Supplementary file 2. Oligo sequences.
- Supplementary file 3. Key resources table.
- Transparent reporting form

## Data availability

Data generated or analyzed during this study are included in the manuscript and supporting files. We did not generate any major datasets such as microarray or DNA sequence data; therefore we no source datasets to provide.

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
