## [Decision Letter]

**Acceptance summary:**

This study provides fundamental insight into the role of cyclophilins in hepatitis C virus replication and infection. The impact of this study is a deeper understanding of how cyclophilins function as co-factors during viral replication, and how cyclophilin inhibitors can confer antiviral effects through a new mechanism involving a cell intrinsic antiviral pathway.

**Decision letter after peer review:**

Thank you for submitting your article "Hepatitis C virus exploits cyclophilin A to evade PKR" for consideration by *eLife*. Your article has been reviewed by three peer reviewers, one of whom is a member of our Board of Reviewing Editors, and the evaluation has been overseen by Päivi Ojala as the Senior Editor. The following individual involved in review of your submission has agreed to reveal their identity: Thomas Pietschmann (Reviewer #2).

The reviewers have discussed the reviews with one another and the Reviewing Editor has drafted this decision to help you prepare a revised submission.

Summary:

In this study, Colpitts et al. provide a broad and molecularly deep study analyzing the role of cyclophilins in hepatitis C virus replication and infection. They used cyclophilin inhibitors to provide evidence that hepatitis C virus co-opts cyclophilin A in order to evade an antiviral restriction pathway dominated by a PKR-IRF1 pathway. They provide additional data indicating that the observed phenotype is independent of RLH/MAVS-induced interferon pathways. The impact of this study is a deeper understanding of how cyclophilins function as co-factors during viral replication, and how Cyp inhibitors can confer antiviral effects through a new mechanism involving a cell intrinsic antiviral pathway.

Essential revisions:

Below are compiled comments from the three reviewers, each one numbered by general topic. The reviewers have requested that additional experiments be performed to address #1-3, and possibly #4, whereas point #5 can likely be addressed without new experimentation.

1) Studies related to further characterizing PKR KO phenotype:

a) (Reviewer #1) The Huh7 PKR KO phenotype with respect to viral replication is quite modest. If PKR is a major contributor to the permissivity differences between Huh7 and Huh7.5, then KO of PKR in Huh7.5 cells should have little effect on virus. This would be a good control to include. Along these lines, what is the impact of PKR KO on virus production? A concern is that these modest effects on HCV-luciferase RLU are not likely to have a major effect on virus production. Lastly, can the PKR KO effect be re-constituted with CRISPR-resistant PKR? Does ectopic expression of PKR in Huh7.5 cells make them more Huh7-like?

b) (Reviewer #2) The strongest and most direct piece of evidence supporting the conclusion that HCV uses CypA to evade PKR-dependent antiviral defences rests on Figure 6 and the use of two PKR knock out clones. The presented data show that HCV replication in the two PKR knock out clones is somewhat more resistant to inhibition by CypIs. Concomitantly, the authors show that mRNA induction of IFN-β by HCV is blunted in the PKR KO clones. In the same figure the authors show that a PKR inhibitor decreases the potency of a CypI in parental Huh7 cells but not so much in Huh-7.5 cells (in a previous figure the authors had shown a greater dependence of HCV on CypA in Huh7 compared to Huh-7.5 cells).

Given the high importance of these data for the main conclusion of the manuscript, in my view the authors should consider cell clonal effects, which potentially could confound this important observation. To this end, the authors could restore PKR expression in the knock out clones. One would expect restoration of HCV sensitivity towards CypIs as well as restoration of antiviral signalling. This would lend further support to the role of CypA in modulation PKR functioning and in turn HCV replication.

Alternatively/in addition, the authors could over-express PKR in Huh7 cells and explore if this enhances sensitivity to CypIs and also antiviral signalling. Taken together such experiments will extend the foundation of this important conclusion.

Along similar lines, the experiment displayed in Figure 6G should also include the comparison between Huh7 cells and the Huh7 PKR knock out clones. In support of the model proposed by the authors, one would think that the PKR inhibitor only shifts HCV sensitivity to CypIs in the Huh7 cells expressing functional PKR but not in the knock out clones. Again, this enhances the experimental foundation for the key conclusion of this paper.

Related to the same Figure 6, the authors should explore if the PKR knock out clones selectively display altered HCV susceptibility to CypIs and not globally to antivirals targeting HCV by different mechanisms (e.g. protease inhibitor, NS5A inhibitor, polymerase inhibitor). While a broad spectrum of different CypIs is already shown in the figure, the authors should consider adding data showing other HCV antivirals.

2) Studies related to the conclusion that RLR-MAVS-IFN does not contribute to phenotype:

a) (Reviewer #1) Much of Figure 3 is dedicated to showing that induction of IFN-β correlates to CypI potency in Huh7, but not Huh7.5 cells. In Figure 6, the authors state that IFN-β induction in the presence of CypI is mediated by PKR. Then in Figure 7, the authors make the case that the IFN-β induction is actually not important (based on ruxolitinib experiments), but that antiviral gene expression through a PKR-IRF1 axis is the key pathway.

The layout of this story was a bit confusing. If IFN-β ends up having no role in the antiviral activity of CypI, why not put this out there up front in Figure 3? This would make the story more clear. That said, the authors should solidify this claim with IFNAR or STAT1 KO cells as genetic proof that IFN-β has no role. Testing IFN-dependent, but IRF1-independent ISG induction would also be a compelling control. Further, if IFN-β has no role here, then IFN bioassays (i.e., transferring soups and testing for inhibition of IFN-sensitive virus like VSV) would further bolster the claim.

b) (Reviewer #2) Exclusion that RLR/MAVS-dependent antiviral responses do not play a role is one major conclusion of this study. Therefore, it seems worthwhile to confirm the unresponsiveness of the cells employed here (and published previously) to RLR/MAVS-dependent viral danger signals.

c) (Reviewer #3) The authors provide data suggesting that CypI induced antiviral signaling is not dependent on RIGI and MAVS. Important controls would be to show that known RLH agonist actually trigger this pathway in their cells, i.e. the cells are signaling competent.

3) Studies related to HCV genotype:

(Reviewer #3) It has been previously been shown that non-JFH1 gts (e.g. Con1) maybe more sensitive to CypA depletion even in Huh7.5 cells. Thus, it would be important to confirm some of the critical data presented here with non-JFH1 based replicons.

4) Concern about the claim that this mechanism is broadly applicable to other viruses:

(Reviewer #3) The authors argue that one of the premises of their study was to investigate the role of CypA for other non-HIV viruses. HCV is arguably a very good example as its dependency on CypA is well established. However, based on the data presented here it is unclear how generalizable CypA-targeted therapies that affect host intrinsic antiviral responses to combat other (RNA) virus infections are. Either additional data are presented to show effects on other viruses or the statements in the manuscript (e.g. Introduction, last paragraph) should be limited to HCV.

5) Additional major concerns that may be addressable without new experimentation:

a) (Reviewer #2) I was somewhat confused by the authors Discussion related to the differential relevance of CypB and A for HCV infection (subsection “Clarifying the roles of Cyps in HCV replication”). In brief, based on which evidence can the authors conclude that CypA is only needed to evade antiviral defences and not directly in replication? After all, both in the PKR and IRF1 knock out clones CypIs still very powerfully suppress HCV replication (albeit in a somewhat higher dose range). One possibility is that here the role of CypA in supporting RNA replication is blocked, thus causing an antiviral effect. In my view it is very difficult to exclude this in the absence of information as to how specifically the CypIs block CypA versus CypB.

b) (Reviewer #3) This reviewer is somewhat surprised by the rather marginal effects of CypA KD in Huh7.5 cells on HCV replication. Previous studies e.g. in HCV-adapted Huh7-Lunet (Kaul et al., 2009) or Huh7.5 cells (Gaska et al., 2019 showed) a 100-500 fold decreased in HCV's ability to replicate. How do the authors reconcile these data with their own?

---

## [Author Response]

Essential revisions:Below are compiled comments from the three reviewers, each one numbered by general topic. The reviewers have requested that additional experiments be performed to address #1-3, and possibly #4, whereas point #5 can likely be addressed without new experimentation.1) Studies related to further characterizing PKR KO phenotype:a) (Reviewer #1) The Huh7 PKR KO phenotype with respect to viral replication is quite modest. If PKR is a major contributor to the permissivity differences between Huh7 and Huh7.5, then KO of PKR in Huh7.5 cells should have little effect on virus. This would be a good control to include. Along these lines, what is the impact of PKR KO on virus production? A concern is that these modest effects on HCV-luciferase RLU are not likely to have a major effect on virus production. Lastly, can the PKR KO effect be re-constituted with CRISPR-resistant PKR? Does ectopic expression of PKR in Huh7.5 cells make them more Huh7-like?

We thank the reviewer for this comment. Unfortunately, we did not manage to knock out PKR in Huh7.5 cells before the lab was closed due to COVID-19. As the reviewer suggests, we do not expect this to make a difference to HCV replication but we have not managed to test this. We have therefore not been able to further study the role of PKR in distinguishing HCV permissivity of Huh7 and 7.5. We believe that this omission does not impact the conclusions in the manuscript. We have re-iterated that the differences between Huh7 and 7.5 cells are multifactorial with the following text:

“Furthermore, our results suggest that a deficiency in PKR-dependent responses, as well as defective RIG-I (Sumpter et al., 2005), may contribute to Huh7.5 cell permissivity for HCV replication, which is consistent with previous observations showing that RIG-I does not play a role (Binder et al., 2007). Indeed, differences in permissivity between Huh7 and Huh7.5 cells are likely governed by a combination of factors, including RIG-I (Sumpter et al., 2005) and CD81 (Koutsoudakis et al., 2007), among others.”

We did over-express PKR in Huh7.5 cells, which inhibited HCV replication. However, we observed that overexpressed PKR was highly active (autophosphorylated) even in the absence of a stimulus, which does not accurately recapitulate the situation in unmodified cells (e.g. Huh7 cells, Figure 7—figure supplement 2A), where PKR is normally inactive (unphosphorylated) until a stimulus is added. Thus, we have not included these particular data in the manuscript as we believe this would complicate the story, without affecting our conclusions about the CypI mechanism. Moreover, these data are similar to those we obtain in the Huh7 PKR KO cells (Author response image 1), which we have included. We have also discussed the issue of over-expressed PKR being constitutively active as relates to these data.

**Author response image 1. sa2fig1:** Huh7 Cells were transduced with lentiviral vector encoding PKR (pSCRPSY-EIF2AK2); transduced cells were selected by addition of puromycin. Expression and activation of PKR was evaluated by Western blot. HCV replication was evaluated as described in the manuscript. HCV replication was slightly decreased in PKR overexpressing Huh7.5 cells. Statistical significance was evaluated by t-test using GraphPad Prism (**** p-value < 0.0001).

b) (Reviewer #2) The strongest and most direct piece of evidence supporting the conclusion that HCV uses CypA to evade PKR-dependent antiviral defences rests on Figure 6 and the use of two PKR knock out clones. The presented data show that HCV replication in the two PKR knock out clones is somewhat more resistant to inhibition by CypIs. Concomitantly, the authors show that mRNA induction of IFN-β by HCV is blunted in the PKR KO clones. In the same figure the authors show that a PKR inhibitor decreases the potency of a CypI in parental Huh7 cells but not so much in Huh-7.5 cells (in a previous figure the authors had shown a greater dependence of HCV on CypA in Huh7 compared to Huh-7.5 cells).Given the high importance of these data for the main conclusion of the manuscript, in my view the authors should consider cell clonal effects, which potentially could confound this important observation. To this end, the authors could restore PKR expression in the knock out clones. One would expect restoration of HCV sensitivity towards CypIs as well as restoration of antiviral signalling. This would lend further support to the role of CypA in modulation PKR functioning and in turn HCV replication.

We thank the reviewer for this important comment. The question of re-expression of PKR in the knockout lines is one we had considered. In the original study, we did not test this because we expected that PKR over-expression would induce translation shutdown as has been described previously (e.g. Grolleau et al., 2000), and would confound our results. Nonetheless, we have now stably over-expressed PKR in Huh7 PKR KO cells and evaluated HCV replication and its sensitivity to CsA inhibition. Over-expressed PKR was highly active, as shown by its robust autophosphorylation even in the absence of a stimulus (Western blot in new Figure 7—figure supplement 2). We found that PKR expression had an inhibitory effect on HCV replication, possibly through suppression of host and viral translation (as evidenced by the autophosphorylation of over-expressed PKR). For this reason, this experiment does not strictly recapitulate the situation in Huh7 cells. That is, the over-expressed PKR is already phosphorylated and activated (see Figure 7—figure supplement 2) instead of un-phosphorylated as in Huh7 cells. Therefore, while PKR over-expression reduced HCV replication, it did not affect HCV sensitivity to CsA. We propose that over-expressed PKR cannot enhance CsA sensitivity, because PKR is already activated and suppressing HCV replication independently of CsA-induced antiviral signaling. We had hoped to examine this further, but did not have time before the COVID-19 closure of the lab.

These data are now shown in Figure 7—figure supplement 2, and we have added the following text:

“We also sought to confirm the role of PKR in CypI activity by over-expressing PKR in Huh7 PKR knockout cells. However, PKR over-expression in itself led to PKR activation, as evidenced by its autophosphorylation (Figure 7—figure supplement 2A). […] This is consistent with previous observations of translation shutdown on PKR overexpression (Grolleau et al., 2000; Barber et al., 1993; Chong et al., 1992; Thomis and Samuel, 1992).”

In the manuscript (originally Figure 6D, now Figure 7D), we observed highly similar results in two PKR KO clones, which were prepared by single-cell cloning. Furthermore, treatment of cells with a PKR inhibitor (originally Figure 6G, now Figure 7G) led to the same phenotype. We also have more new C16 data supporting a role for PKR (Figure 7G). Thus, while the PKR reconstitution experiment did not yield clear results, we propose that together, these data are sufficiently compelling.

Alternatively/in addition, the authors could over-express PKR in Huh7 cells and explore if this enhances sensitivity to CypIs and also antiviral signalling. Taken together such experiments will extend the foundation of this important conclusion.

As described above, PKR over-expression does not accurately recapitulate the situation in naïve cells in that overexpressed PKR is highly active (autophosphorylated) even in the absence of a stimulus. This makes it difficult to evaluate the effect on the sensitivity of HCV to CypI, which we propose relies on more subtle PKR-mediated activation of IRF1, in the absence of translation shutoff. We have instead focused on an alternative approach to provide further evidence for a role for PKR using the PKR inhibitor C16 (Figure 7G).

Along similar lines, the experiment displayed in Figure 6G should also include the comparison between Huh7 cells and the Huh7 PKR knock out clones. In support of the model proposed by the authors, one would think that the PKR inhibitor only shifts HCV sensitivity to CypIs in the Huh7 cells expressing functional PKR but not in the knock out clones. Again, this enhances the experimental foundation for the key conclusion of this paper.

This is a very helpful suggestion. We have now compared the effect of the C16 PKR inhibitor in Huh7 and Huh7 knocked out for PKR. The result was as expected and PKR knockout prevents the effect of C16 on CsA sensitivity.

These new data are included as Figure 7G. We have added the following to the text:

“C16 decreased CsA potency in Huh7 cells, but not in Huh7 PKR KO cells (Figure 7G), and only minimally affected CsA potency against HCV in Huh7.5 cells (Figure 7—figure supplement 1).”

Related to the same Figure 6, the authors should explore if the PKR knock out clones selectively display altered HCV susceptibility to CypIs and not globally to antivirals targeting HCV by different mechanisms (e.g. protease inhibitor, NS5A inhibitor, polymerase inhibitor). While a broad spectrum of different CypIs is already shown in the figure, the authors should consider adding data showing other HCV antivirals.

This is another important point. We have now evaluated the antiviral activities of daclatasvir (NS5A inhibitor) and telaprevir (NS3/4A protease inhibitor) in Huh7 and Huh7 PKR KO cells. As expected, the presence or absence of PKR did not affect HCV sensitivity to these other inhibitors. These data are now shown in Figure 7—figure supplement 3. We have added the following text:

“We next tested whether the absence of PKR broadly affects the sensitivity of HCV to the antiviral activity of telaprevir (NS3/4A protease inhibitor) and daclatasvir (NS5A inhibitor) in Huh7 and Huh7 PKR KO cells. […] Collectively, these data are consistent with a specific role for PKR in the enhanced antiviral activity of CypI in Huh7 cells.”

2) Studies related to the conclusion that RLR-MAVS-IFN does not contribute to phenotype:a) (Reviewer #1) Much of Figure 3 is dedicated to showing that induction of IFN-β correlates to CypI potency in Huh7, but not Huh7.5 cells. In Figure 6, the authors state that IFN-β induction in the presence of CypI is mediated by PKR. Then in Figure 7, the authors make the case that the IFN-β induction is actually not important (based on ruxolitinib experiments), but that antiviral gene expression through a PKR-IRF1 axis is the key pathway.The layout of this story was a bit confusing. If IFN-β ends up having no role in the antiviral activity of CypI, why not put this out there up front in Figure 3? This would make the story more clear. That said, the authors should solidify this claim with IFNAR or STAT1 KO cells as genetic proof that IFN-β has no role. Testing IFN-dependent, but IRF1-independent ISG induction would also be a compelling control. Further, if IFN-β has no role here, then IFN bioassays (i.e., transferring soups and testing for inhibition of IFN-sensitive virus like VSV) would further bolster the claim.

We have now re-structured the manuscript to present the IFN-β/ruxolitinib data earlier, in what is now Figure 4. We have also sought to clarify that the observed phenotype is not dependent on IFN-β signalling and IFN induced gene expression, but is rather a cell-intrinsic response (i.e., gene expression dependent on PKR-IRF1 activation) with IFN-β being one of many antiviral genes induced upon CypI treatment of Huh7, but not Huh7.5 cells. We now show the activation of other antiviral genes, which are predominantly IRF1-dependent, in Figure 4. To confirm that the effect is not mediated by IFN, we performed further experiments using an interferon receptor (IFNAR) blocking antibody. As for ruxolitinib, IFNAR blockade did not affect the CypI potency against HCV, although it did rescue HCV replication from the inhibitory effect of IFN. This data is presented in Figure 4—figure supplement 3.

We have added the following text:

“The notion that IFN was not required for maximal CypI activity was also supported by an experiment using antihuman interferon alpha/beta receptor chain 2 antibody (IFNAR) to inhibit IFN activity through receptor blockade. […] However, in vivo, IFN induction would be expected to influence HCV replication and adaptive immune responses and thus the antiviral activity of CypI in patients.”

b) (Reviewer #2) Exclusion that RLR/MAVS-dependent antiviral responses do not play a role is one major conclusion of this study. Therefore, it seems worthwhile to confirm the unresponsiveness of the cells employed here (and published previously) to RLR/MAVS-dependent viral danger signals.c) (Reviewer #3) The authors provide data suggesting that CypI induced antiviral signaling is not dependent on RIGI and MAVS. Important controls would be to show that known RLH agonist actually trigger this pathway in their cells, i.e. the cells are signaling competent.

Unfortunately, we did not manage to complete this experiment before our qPCR machine was requisitioned for COVID-19 testing outside UCL. Nonetheless, RLR/MAVS-dependent antiviral responses have previously been documented in Huh7 cells and we have now cited these studies more prominently. We have added new text reiterating this point:

“Huh7 cells are capable of responding to cytosolic RNA and initiating antiviral signalling through RIG-I, MAVS and IRF3 (Sumpter et al., 2005; Binder et al., 2007). […] Therefore, we first hypothesized that the active RIG-I pathway in Huh7 cells contributes to the antiviral signalling induced by CypI.”

3) Studies related to HCV genotype:(Reviewer #3) It has been previously been shown that non-JFH1 gts (e.g. Con1) maybe more sensitive to CypA depletion even in Huh7.5 cells. Thus, it would be important to confirm some of the critical data presented here with non-JFH1 based replicons.

We thank the reviewer for this suggestion. We attempted to evaluate replication of the Con1 (genotype 1b) replicon (Lohmann et al. 2003 J. Virol. 77: 3007-3019). Despite significant effort, we were unable to get the Con1 replicon to replicate in our Huh7 and Huh7.5 cells. In the past month, we obtained the Huh7-Lunet cell line, which better supports Con1 replication. We were able to observe weak replication in Huh7-Lunet cells, but ran out of time before the COVID-19 lab closure and were unable to evaluate the effect of CypA depletion or CypI treatment in the presence or absence of PKR in the Huh7-Lunet cell line.

Importantly, Kaul et al., 2009, evaluated replication of the Con1 replicon in CypA-depleted Huh7-Lunet cells and Huh7.5 cells. CypA depletion in Huh7-Lunet cells profoundly inhibited HCV replication by approximately 3 logs at 48 hours, and had a lesser impact in Huh7.5 cells (Kaul et al., 2009, Figure 2B). These data suggest that our findings are relevant for other HCV genotypes and we now cite this study to make this point in the Discussion as follows:

“Notably, a differential requirement for CypA in Huh7-Lunet and Huh7.5 cells has also been observed for replication of genotype 1b (Con1) and genotype 2a (JFH-1) replicons (Kaul et al., 2009), suggesting that these mechanisms are consistent across HCV genotypes.”

4) Concern about the claim that this mechanism is broadly applicable to other viruses:(Reviewer #3) The authors argue that one of the premises of their study was to investigate the role of CypA for other non-HIV viruses. HCV is arguably a very good example as its dependency on CypA is well established. However, based on the data presented here it is unclear how generalizable CypA-targeted therapies that affect host intrinsic antiviral responses to combat other (RNA) virus infections are. Either additional data are presented to show effects on other viruses or the statements in the manuscript (e.g. Introduction, last paragraph) should be limited to HCV.

We agree with the reviewer that we had over-generalized our conclusions and in fact the mechanisms we have identified here may or may not be applicable to other viruses. To address this concern, we have limited our discussion in the current manuscript to HCV, with the exception of where we have cited literature describing other viruses and CypA.

5) Additional major concerns that may be addressable without new experimentation:a) (Reviewer #2) I was somewhat confused by the authors Discussion related to the differential relevance of CypB and A for HCV infection (subsection “Clarifying the roles of Cyps in HCV replication”). In brief, based on which evidence can the authors conclude that CypA is only needed to evade antiviral defences and not directly in replication? After all, both in the PKR and IRF1 knock out clones CypIs still very powerfully suppress HCV replication (albeit in a somewhat higher dose range). One possibility is that here the role of CypA in supporting RNA replication is blocked, thus causing an antiviral effect. In my view it is very difficult to exclude this in the absence of information as to how specifically the CypIs block CypA versus CypB.

We thank the reviewer for pointing out this lack of clarity in our Discussion. This was partly due to a typographical error where we had mixed up CypA and CypB in the original Discussion. This is now fixed in the new text. To further address this point, we have added a new panel showing that CypA depletion does not impact CsA activity over a CsA dose range in Huh7.5 cells (Figure 1—figure supplement 2), rather than just a single dose as we had shown previously (Figure 1E). We have also clarified the Discussion as follows:

“Here, our data support a direct role for CypB in HCV RNA replication (Watashi et al., 2005), which is consistent with it being equally required in Huh7 and Huh7.5 cells (Figure 1C). In contrast, the requirement for CypA varies according to cell line and appears to be important for evasion of host antiviral responses in innate sensing competent cells (i.e., Huh7) (Figure 1C).”

We have referred to the new data as follows:

“CsA also inhibited HCV replication in Huh7.5 cells silenced for CypA expression (Figure 1E), with similar antiviral potency regardless of CypA expression (Figure 1—figure supplement 2).

b) (Reviewer #3) This reviewer is somewhat surprised by the rather marginal effects of cypA KD in Huh7.5 cells on HCV replication. Previous studies e.g. in HCV-adapted Huh7-Lunet (Kaul et al., 2009) or Huh7.5 cells (Gaska et al., 2019 showed) a 100-500 fold decreased in HCV's ability to replicate. How do the authors reconcile these data with their own?

We thank the reviewer for pointing out this oversight in neglecting to include this in our Discussion. Gaska et al., 2019, evaluated the effect of CypA knockdown using HCVcc infection models. CypA has been suggested to be involved in additional steps of HCV infection, such as assembly, which would not have an effect in our replicon assay. In our HCVcc assays, CypA depletion in Huh7.5 cells decreased HCVcc infection by 1-2 logs (Figure 1F), which is consistent with what has been observed in the literature. We have now added a paragraph in the Discussion to reconcile this data with our own:

“CypA has been proposed to have a role in HCV assembly (Nag et al., 2012; Anderson et al., 2011), which is likely reflected by our observation that CypA depletion in Huh7.5 cells decreased HCVcc infection (Figure 1F), but not replication of the HCV replicon (Figure 1C). This is consistent with previous studies showing a large decrease in HCVcc infection in Huh7.5 cells silenced for CypA expression (Gaska et al., 2019).”

With regards to the study by Kaul et al., 2009), only a small decrease in HCV replication in CypA-depleted Huh7.5 cells was observed at 48 hours (Kaul et al., Figure 2A). This is consistent with our data, in that we do see a ~2-fold decrease in HCV replication in Huh7.5 CypA knockdown cells. We have replotted the data on a linear scale and have shown it in Author response image 2 to highlight this point. Kaul et al. observed a more pronounced defect in HCV replication at 48 hours in CypA-depleted Huh7-Lunet cells (Kaul et al. Figure 2A), which is consistent with our observations as well. We have added a sentence to our Discussion to support this:

“Notably, a differential requirement for CypA in Huh7-Lunet and Huh7.5 cells has also been observed for replication of genotype 1b (Con1) replicons and genotype 2a replicons (Kaul et al., 2009), suggesting that these mechanisms are consistent across HCV genotypes.”

**Author response image 2. sa2fig2:** HCV replication in CypA-depleted Huh7.5 cells.